# Multimodal Image Registration Guided by Few Segmentations from One Modality

**Başar Demir**                                  BDEMIR@CS.UNC.EDU
**Marc Niethammer**                            MN@CS.UNC.EDU
*University of North Carolina at Chapel Hill, North Carolina, USA*

**Editors:** Accepted for publication at MIDL 2024

## Abstract

Registration of multimodal images is challenging, especially when dealing with different anatomical structures and samples without segmentations. The main difficulty arises from the use of registration loss functions that are inadequate in the absence of corresponding regions. In this work, we present the first registration and segmentation approach tailored to this challenge. In particular, we assume the practically highly relevant scenario that only a limited number of segmentations are available for one modality and none for the other. First, we augment our few segmented samples using unsupervised deep registration within one modality, thereby providing many anatomically plausible samples to train a segmentation network. The resulting segmentation network then allows us to train a segmentation network on the target modality without available segmentations by using an unsupervised domain adaptation architecture. Finally, we train a deep registration network to register multimodal image pairs purely based on predictions of their segmentation networks. Our work demonstrates that using a small number of segmentations from one modality enables training a segmentation network on a target modality without the need for additional manual segmentations on that modality. Additionally, we show that registration based on these segmentations provides smooth and accurate deformation fields on anatomically different image pairs, unlike previous methods. We evaluate our approach on 2D medical image segmentation and registration between knee DXA and X-ray images. Our experiments show that our approach outperforms existing methods. *Code is available at* https://github.com/uncbiag/SegGuidedMMReg.

**Keywords:** multimodal image registration, few-shot learning

## 1. Introduction

Image registration is a crucial medical imaging task in establishing correspondences between image pairs. It is commonly used for disease diagnosis (Khalil et al., 2018), disease progression monitoring (Viergever et al., 2016), and organ motion tracking (Fu et al., 2020). At its core is the solution of an inverse problem to determine the unknown spatial transformation between a moving and a fixed image, such that the deformed moving image matches the fixed image well. Solving this inverse problem involves solving a possibly high-dimensional optimization problem. One can *directly* optimize over the parameters of a chosen transformation model (e.g., a displacement vector field) or over the parameters of a deep neural network which then *indirectly* produces the desired transformation by regression. Deeplearning approaches (Balakrishnan et al., 2019; Yang et al., 2017) are typically significantly faster than direct optimization approaches (Avants et al., 2009; Heinrich et al., 2014) and

can achieve state-of-the-art (SOTA) accuracies ([Tian et al., 2023](#); [Mok and Chung, 2020](#)) especially when combined with instance optimization.

All of these methods aim to optimize a loss function that is a compromise between image pair dissimilarity and transformation irregularity. Multimodal image registration is still significantly more challenging than monomodal image registration because it requires image dissimilarity costs that can assess whether two images from different modalities differ or should be considered spatially well-matched. Further, Fig. [1](#) illustrates that the visibility of different anatomical structures across different modalities may cause a registration model to predict incorrect correspondences between different anatomical structures. *We hypothesize that a registration model trained using supervision based on segmentations available for both modalities can assist the registration model in focusing on common regions of image pairs.* However, this presupposes the availability of these segmentations. Our focus in this work is, therefore, to develop an approach that solves both image segmentation and registration for multimodal image pairs, assuming that 1) anatomies are not consistently visualized between the modalities (e.g., one might focus on bone visualization while another might visualize both bones and soft tissue) and 2) only a few manual segmentations are available in one modality and none in the other.

**In summary, the contributions of our work are:**

- We propose the first deep image registration approach for multimodal images where image pairs visualize different anatomical parts. Our approach is based on registering segmentations while only requiring a few segmentations from one modality and none for the other.

- We test our approach for 2D knee datasets between DXA images which focus on imaging bone, and radiographs (termed X-rays in what follows) which visualize bones and soft tissue.

- Compared to existing loss functions and approaches, our approach can reliably register DXA and X-ray image pairs and predicts smooth deformation fields on anatomically different image pairs.

## 2. Related Work

**Image Registration.** Image registration is a key tool of medical image analysis to estimate spatial correspondences between images. Recently, deep neural networks have enabled population-based registration approaches that require one-time training but then provide very fast inference ([Balakrishnan et al., 2019](#); [Miao et al., 2016](#); [Greer et al., 2021](#); [Tian et al., 2023](#); [Greer et al., 2023](#)). However, registration performance still heavily depends on selecting a correct task-specific image similarity measure.

For monomodal applications, mean squared error (MSE) ([Fan et al., 2019](#); [Hoopes et al., 2021](#); [Kim et al., 2021](#)) and normalized cross-correlation (NCC) ([Kim et al., 2022](#); [Tang et al., 2020](#)) are popular image similarity measures. In contrast, multimodal registration cannot rely on matching image intensities directly. Hence, measures of statistical dependence such as NCC and Mutual Information ([Wells III et al., 1996](#)) are popular choices. Another option, Modality Independent Neighbourhood Descriptors (MIND) ([Heinrich et al.,](#)

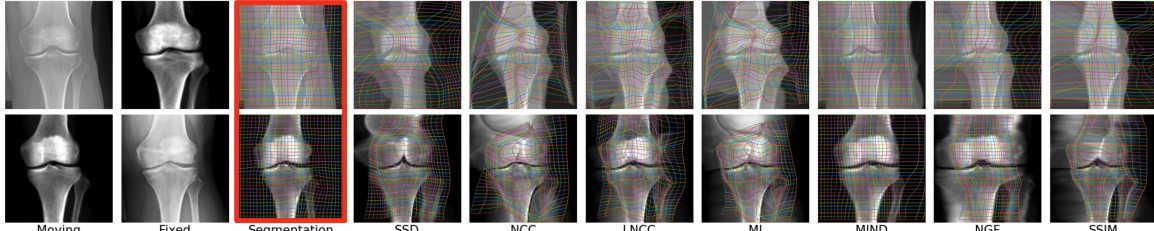

Figure 1: Multimodal registration results for registration using different similarity measures. Existing multimodal registration approaches based on image intensities fail to capture anatomical correspondences, whereas our segmentation-based approach (see red box) predicts smooth and accurate deformation fields. *DXA images reproduced by kind permission of the UK Biobank®.*

2012), operates by comparing local image neighborhoods in both the moving and fixed images. Many other image similarity measures exist (e.g., structural similarity index (SSIM) (Wang et al., 2004) and normalized gradient fields (NGF) (Haber and Modersitzki, 2006)). However, none of these similarity measures are usable for image pairs which show different anatomical parts (see Fig. 1). Specifically, registrations based on these similarity measures fail to find appropriate correspondences even between common anatomical structures when one of the images contains anatomical structures that are not visualized in the other image.

Recent works (Xu and Niethammer, 2019; Canalini et al., 2019; Cai et al., 2022) show that segmentation can supervise registration and improve registration performance with better border alignment. Further, such supervision encourages segmentations to match after registration, ensuring the preservation of anatomical consistency. Inspired by these works, we propose a *registration strategy that can find correspondences fully based on given segmentations tailored for anatomically different and multimodal image pairs.*

**Domain Adaptation for Segmentation.** Machine learning models often struggle with data that differs significantly from their training data. This is a major challenge in medical imaging, where datasets can vary greatly in modality (e.g., MRI vs. CT) even when imaging the same body region. Moreover, manual segmentations are typically available for only a limited number of images and modalities. Unsupervised domain adaptation (UDA) offers a solution to this challenge. UDA techniques adapt models trained on one domain (source) to perform well on a related but different domain (target) without requiring segmented target data. This is crucial for generalizing models in medical imaging, where acquiring large segmented datasets is expensive and time-consuming. Building on the success of adversarial image translation methods (Liu and Tuzel, 2016; Isola et al., 2017; Zhu et al., 2017), several techniques have been proposed for pixel-level UDA (Murez et al., 2018; Pizzati et al., 2020; Li et al., 2020). These methods initially establish a mapping function between the source and target domains, enabling inter-domain image translation.

Recent works (Hoffman et al., 2018; Xie et al., 2018) enhance UDA by combining feature alignment and pixel-level adversarial training. In our approach, we use an *CycleGAN (Zhu et al., 2017) unsupervised domain adaptation strategy* with semantic losses *at each step of*

*translation (Gogoll et al., 2020).* These semantic losses ensure that modality-translated images not only appear to belong to the target image domain but also maintain consistent anatomical boundaries during image translation. This is an essential property for our registration approach. *Further, the approach allows us to train a segmentation network on the target modality using only source domain segmentations.*

## 3. Methodology

### 3.1. Overview

Our source dataset $D_S = \{(I_i)\}_{i \in [n]}$ includes at least one segmentation $\{(S_i)\}_{i \in [l]}$ ($1 \leq l \leq n$). No segmentations are available for our target dataset $D_T = \{(I_i)\}_{i \in [m]}$. First, we augment the source dataset $D_S$ with a registration network (Sec. 3.2). Subsequently, we train the segmentation network $S_S$ for the source domain using the augmented dataset and freeze weights of the network. Next, we obtain a segmentation network $S_T$ for the target dataset with the guidance of $S_S$ via an unsupervised domain adaptation network (Sec. 3.3). We predict segmentations for the non-segmented part of the source dataset with $S_S$ and apply $S_T$ to the target dataset. Finally, we train a registration network solely based on the segmentations we obtained (Sec. 3.4). This registration network then enables us to register two images according to their predicted or provided segmentations.

### 3.2. Augmentation with Registration

A non-parametric registration network predicts a displacement field $u = R(I_m, I_f; \theta)$ between the moving image $I_m$ and the fixed image $I_f$ for given network parameters $\theta$. We can warp the moving image to the fixed image as $I_m^w = I_m \circ \Phi^{-1}$, where $\Phi^{-1} = u + id$ is a transformation map, and $id$ is the identity transform. Since the registration network can find correspondences between images without supervision, we use predicted transformation maps to create artificial samples by warping the images with segmentations to images without available segmentations. First, we sample moving image segmentation pair $(I_m, S_m) \in D_S$ and fixed image $I_f \in D_S$ and we obtain the transformation map $\Phi^{-1}$ between $I_m$ and $I_f$. Next, we create a new sample by warping both the moving image $I_m$ *and* its segmentation $S_m$ as

$$(I_m^w, S_m^w) = \left(I_m \circ \Phi^{-1}, S_m \circ \Phi^{-1}\right) . \tag{1}$$

Warping both the image and its segmentation helps preserve semantic consistency between them. This artificially created sample, $(I_m^w, S_m^w)$, is anatomically similar to $I_f$. When a limited number of segmented samples is available, we can use this approach to create additional samples. Sec. B.2 provides details of the implementation.

### 3.3. Domain Adaptation

Inspired by (Gogoll et al., 2020), we use a CycleGAN (Zhu et al., 2017)-based domain adaptation network for medical image *segmentation*. We assume that we already have a pretrained segmentation network $S_S$ for the source domain. Our goal is to train a segmentation network $S_T$ for the target domain by transferring information from $S_S$.

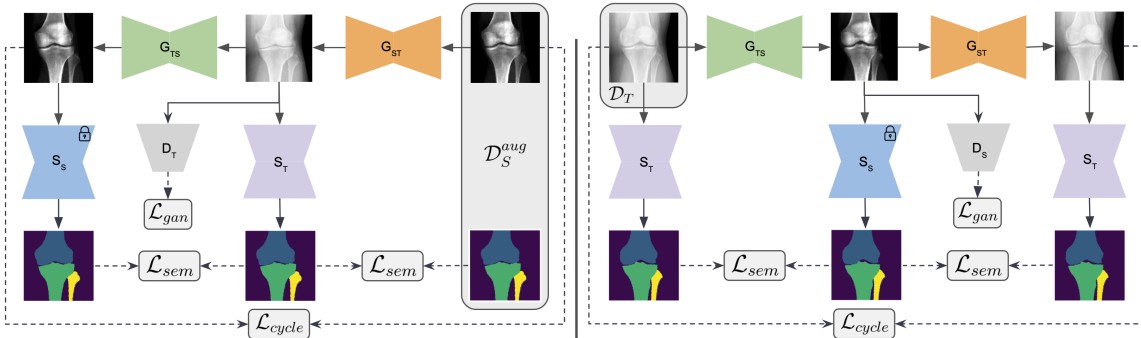

Figure 2: **Domain Adaptation Overview.** Our domain adaptation architecture consists of two image translation cycles. Each cycle translates images from one domain (source or target) to the other and back again. Our pre-trained segmentation network $S_S$ enables training of $S_T$ based on semantic consistency. *DXA images reproduced by kind permission of the UK Biobank®.*

We use an adversarial training procedure. Two generators, $G_{ST}$ and $G_{TS}$, are designed to learn mapping functions from the source domain to the target domain and vice versa. Concurrently, discriminators $D_S$ and $D_T$ are responsible for distinguishing between real and translated samples. We integrate these generators and discriminators into the image translation process with adversarial loss ($\ell_{GAN}$) and cycle consistency loss ($\ell_{\text{cycle}}$) terms, following the principles of CycleGAN (Zhu et al., 2017). However, it is important to note that this process does not ensure the semantic consistency between real and translated images, i.e., while translated images look realistic for a CycleGAN approach the synthesized image will not necessarily be spatially consistent, and anatomical boundaries might shift. This would, of course, be problematic for any subsequent registration. Hence, we incorporate semantic segmentation networks $S_S$ and $S_T$ to encourage the semantic segmentation of the images at each step of the cycle to remain consistent (Gogoll et al., 2020). The work by (Gogoll et al., 2020) assumes that all segmentations are available for the source domain, and hence manual segmentations can be used for network training. However, in our case, only a few segmentations are available for the source domain. Therefore, we augment our source dataset with registration (see Sec. 3.2), and use all images in $D_S$ for training.

To encourage consistent segmentation predictions at each cycle phase we use the losses

$$\ell_{\text{sem-ST}} := \mathbb{E}_{(I_S, M_S) \sim \mathcal{D}_S^{aug}} \mathcal{L}_{sem}\left(S_T\left(G_{ST}(I_S)\right), M_S\right) + \mathcal{L}_{sem}\left(S_S\left(G_{TS}\left(G_{ST}(I_S)\right)\right), M_S\right),$$
$$\ell_{\text{sem-TS}} := \mathbb{E}_{I_T \sim \mathcal{D}_T} \mathcal{L}_{sem}\left(S_T(I_T), S_S\left(G_{TS}(I_T)\right)\right) + \mathcal{L}_{sem}\left(S_T(I_T), S_T\left(G_{ST}\left(G_{TS}(I_T)\right)\right)\right) \tag{2}$$

where $\mathcal{D}_S^{aug}$ is augmented source dataset containing image segmentation pairs $(I_S, M_S)$. These losses enable us to train $S_T$ based on $S_S$ which was trained with only a few segmented samples. The total loss is

$$\ell_{\text{total}} = \lambda_{GAN}\left(\ell_{GAN}^{ST} + \ell_{GAN}^{TS}\right) + \lambda_{cycle}\ell_{\text{cycle}} + \lambda_{sem}\left(\ell_{\text{sem-ST}} + \ell_{\text{sem-TS}}\right). \tag{3}$$

In summary, we use an unsupervised domain adaptation method for medical image segmentation (Gogoll et al., 2020). In doing so, we jointly learn image segmentation and translation using a pre-trained segmentation network trained with only a few segmentations. Sec. B.3 provides implementation details.

### 3.4. Registration by Segmentation

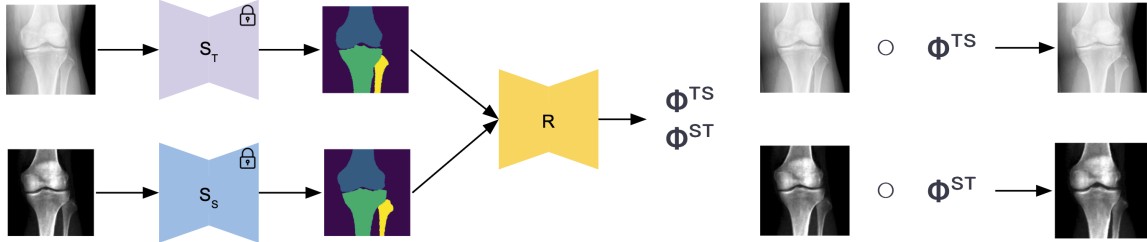

Figure 3: **Registration Approach.** We first predict segmentations of the images by our $S_S$ and $S_T$ segmentation models. Then, we pass the segmentations to the registration network and predict transformation maps. Finally, we warp images based on the predicted transformation maps. *DXA images reproduced by kind permission of the UK Biobank®.*

Instead of proposing a new registration approach, we use an inverse-consistent registration network (Tian et al., 2023). Since our goal is to register anatomically different pairs, we register images based on their segmentations. Note that while our domain adaptation approach of Sec. 3.3 generates translated images and segmentations, we only utilize its capability to train a segmentation network for the target domain.

Our image-segmentation pairs for moving and fixed images are $(I_m, S_m)$ and $(I_f, S_f)$. In our registration setting, the registration network $R$ finds correspondences between images based on their segmentations. The registration network predicts the transformation map $\Phi = R(S_m, S_f; \theta) + id$, where $\theta$ denotes the network parameters. We denote by $\Phi_\theta[S_m, S_f]$ the transformation mapping the moving segmentation, $S_m$ to the space of the fixed segmentation, $S_f$. During training, we register segmentations of image pairs, using gradient inverse consistency of the transformation maps as the regularizer and 1-Dice score on the segmentations as the similarity loss $\mathcal{L}_{\mathrm{sim}}$. The training loss function is

$$\ell = \mathcal{L}_{\mathrm{sim}}\left(S_m \circ \Phi_\theta\left[S_m, S_f\right], S_f\right) + \mathcal{L}_{\mathrm{sim}}\left(S_f \circ \Phi_\theta\left[S_f, S_m\right], S_m\right) + \\ + \lambda \left\|\nabla\left(\Phi_\theta\left[S_m, S_f\right] \circ \Phi_\theta\left[S_f, S_m\right]\right) - \mathbf{I}\right\|_F^2. \quad (4)$$

During inference, we first register the segmentation pair $(S_m, S_f)$ and obtain the corresponding transformation map $\Phi_\theta[S_m, S_f]$. Then, we use this transformation map to warp the real multimodal image pair $(I_m, I_f)$ as $I_m^w = I_m \circ \Phi_\theta[S_m, S_f]$.

In summary, we use our pre-trained segmentation networks (see Sec. 3.2, Sec. 3.3) for inference. First, we predict the segmentations for a given image pair. Then, we predict the

transformation map based on the predicted segmentation pair. Lastly, we warp the real images using this predicted transformation map. Thus, we preserve anatomical consistency while finding correspondences between images. Sec. B.4 provides implementation details.

## 4. Experiments

We conduct experiments to evaluate the effectiveness of each step of our approach. We use 123 DXA samples provided by the UK Biobank (Sudlow et al., 2015) and 123 X-rays from The Osteoarthritis Initiative (OAI) (Nevitt et al., 2006)[1]. We resized each sample to $256 \times 256$ and manually segmented the femur, tibia, and fibula for 100 samples from each dataset. We performed affine pre-alignment on all samples to one random sample using the affine layers of (Greer et al., 2023). See Table 2 for the dataset splits.

**Augmentation with Registration:** We compare the effectiveness of our augmentation strategy with raw data (without any augmentation) and elastic augmentation (Ronneberger et al., 2015). Our method outperforms elastic augmentation in most cases. Additionally, we investigate the impact of the number of segmented training samples on test performance. Table 5 (highlights in orange cells) shows that we achieve comparable performance to the segmentation network trained on 75 segmented samples using only 10 segmented samples for DXA (86.6% fewer) or 15 samples for X-rays (80% fewer). Additionally, our method does not require any parameters unlike elastic augmentation (e.g., number of control points and maximum displacement).

**Domain Adaptation for Segmentation:** We evaluate our domain adaptation network in both the DXA to X-ray and the X-ray to DXA adaptation directions. We assume that we already have a pre-trained segmentation model on the source domain trained with varying numbers of segmented samples. We investigate the impact of different ways of creating the source dataset. We conduct experiments using the following approaches: 1) "Baseline," which only uses images with available segmentations; 2) "Baseline + Predictions," which additionally uses pre-trained network predictions for all available images without segmentation; and 3) "Baseline + Augmentation," which augments the source dataset with our augmentation strategy. We observe that augmenting the source dataset is the most reliable approach when the number of segmentations is less than 5. However, for 15 segmentations or more, all approaches converge to similar Dice scores.

The domain adaptation network significantly improves performance on the target modality. The "Baseline + Augmentation" approach improves over directly using a segmentation network trained *only* on the source domain and then applied to the target domain ("w/o Adaptation"). For example, when only 5 segmented samples are available for the source domain, we observe a 7.1% increase in mean Dice score when comparing the "Baseline + Augmentation" and the "w/o Adaptation" approaches when adapting from the X-ray to the DXA domain, and an increase of 50.5% in Dice score when adapting from the DXA to the X-ray domain. Table 6 shows quantitative results, and we highlight the results discussed herein orange cells. Figures 7 and 8 provide visual comparisons.

**Registration:** We compare our approach to deep registration approaches that differ with respect to their input (either images or segmentations) and the losses that were used during

---

1. This registration task is motivated by our desire to extract spatially localized biomarkers across multiple large-scale, but modality-diverse datasets, e.g., to enable imaging genetics studies for large sample sizes.

|  |  |  |  |  | DXA → X-ray | | | | | X-ray → DXA | | | | |  |
| Approach | Input Type | #DXA Seg. | #X-ray Seg. | Loss | Femur | Tibia | Fibula | Mean | $\%\,|J_\varphi|$ | Femur | Tibia | Fibula | Mean | $\%\,|J_\varphi|$ | Mean |
|---|---|---|---|---|---|---|---|---|---|---|---|---|---|---|---|
| 1 | Manual Segmentations | 75 | 75 | Dice | 0.993 | 0.993 | 0.990 | 0.992 | 0.000 | 0.994 | 0.995 | 0.990 | 0.993 | 0.000 | 0.993 |
| 2 | Real DXA - Real X-ray | ✗ | ✗ | LNCC | 0.834 | 0.813 | 0.610 | 0.752 | 0.000 | 0.952 | 0.950 | 0.841 | 0.914 | 0.845 | 0.883 |
| 3 | Real DXA - Real X-ray | 75 | 75 | NGF | 0.966 | 0.963 | 0.869 | 0.933 | 0.000 | 0.964 | 0.960 | 0.865 | 0.930 | 0.000 | 0.931 |
| 4 | Real DXA - Real X-ray | 75 | 75 | Dice | 0.966 | 0.965 | 0.863 | 0.931 | 0.000 | 0.966 | 0.967 | 0.844 | 0.926 | 0.000 | 0.929 |
| 5 | **Real DXA - Fake DXA** | ✗ | ✗ | LNCC | 0.970 | 0.964 | 0.802 | 0.912 | 0.000 | 0.964 | 0.960 | 0.817 | 0.914 | 0.000 | 0.913 |
|  | | 75 | 75 | | 0.969 | 0.967 | 0.920 | 0.952 | 0.000 | 0.954 | 0.958 | 0.908 | 0.940 | 0.000 | 0.946 |
|  | Real X-ray - Fake X-ray | ✗ | ✗ | LNCC | 0.966 | 0.962 | 0.806 | 0.911 | 0.000 | 0.966 | 0.960 | 0.818 | 0.915 | 0.000 | 0.913 |
|  | | 75 | 75 | | 0.961 | 0.960 | 0.866 | 0.929 | 0.000 | 0.966 | 0.962 | 0.880 | 0.936 | 0.000 | 0.933 |
| Ours | **Predicted Segmentations** | 1 | ✗ | Dice | 0.956 | 0.945 | 0.899 | 0.933 | 0.000 | 0.957 | 0.944 | 0.902 | 0.934 | 0.000 | 0.891 |
|  | | 3 | ✗ | Dice | 0.967 | 0.964 | 0.917 | 0.949 | 0.000 | 0.967 | 0.965 | 0.919 | 0.950 | 0.000 | 0.950 |
|  | | 5 | ✗ | **Dice** | **0.970** | **0.966** | **0.932** | **0.956** | **0.000** | **0.971** | **0.967** | **0.934** | **0.957** | **0.000** | **0.957** |
|  | | 10 | ✗ | Dice | 0.972 | 0.972 | 0.925 | 0.956 | 0.000 | 0.972 | 0.973 | 0.928 | 0.958 | 0.000 | 0.957 |
|  | | 75 | ✗ | Dice | 0.970 | 0.970 | 0.938 | 0.959 | 0.000 | 0.971 | 0.971 | 0.939 | 0.960 | 0.000 | 0.960 |
|  | | ✗ | 1 | Dice | 0.917 | 0.942 | 0.793 | 0.884 | 0.002 | 0.906 | 0.946 | 0.825 | 0.892 | 0.008 | 0.888 |
|  | | ✗ | 3 | Dice | 0.971 | 0.965 | 0.901 | 0.946 | 0.000 | 0.969 | 0.967 | 0.916 | 0.951 | 0.001 | 0.948 |
|  | | ✗ | 5 | Dice | 0.966 | 0.965 | 0.889 | 0.940 | 0.000 | 0.966 | 0.965 | 0.905 | 0.945 | 0.000 | 0.943 |
|  | | ✗ | 10 | **Dice** | **0.976** | **0.967** | **0.937** | **0.960** | **0.000** | **0.977** | **0.969** | **0.935** | **0.960** | **0.000** | **0.960** |
|  | | ✗ | 75 | Dice | 0.977 | 0.972 | 0.950 | 0.966 | 0.000 | 0.978 | 0.975 | 0.953 | 0.969 | 0.000 | 0.968 |

Table 1: Dice and % of voxels with neg. Jacobian ($\%\,|J_\varphi|$) for multimodal registration.

training (either using 1-Dice for segmentation-based losses or various image-based similarity measures). Specifically, we compare our registration approach with 1) An oracle approach that has access to manual segmentations for all images and uses manual segmentations as input and for the loss (to provide an upper bound on performance); 2) Multimodal registration where inputs are real images and our loss uses images; 3) Multimodal registration where inputs are real images and our loss uses images but only within manually segmented regions (Sec. C.1); 4) Multimodal registration where inputs are real images and our loss uses manual segmentations (Sec. C.2); 5) Monomodal registration where inputs are real-fake image pairs generated by our image translation network, where loss uses images (Sec. C.3).

Note that while our approach only assumes that few segmentations are available on the source domain, approaches 1, 3, 4, and 5 above may use manual segmentations for all images in the source and target domain to provide strong competing approaches.

Table 1 shows that our approach achieves a mean Dice score of 95.7% with only 5 segmentations from the DXA domain or 96% with 10 segmentations from the X-ray domain, which is better than the non-oracle best-performing approach (5) that uses 150 manual segmentations in the training process and results in a Dice score of 94.6%. For more detailed results, please refer to Table 7 where we compare different approaches with varying training losses. Additionally, Figures 9 and 10 demonstrate that our method aligns bone borders with smooth deformation fields.

## 5. Conclusion

We demonstrated that our proposed approach enables the registration of 2D multimodal image pairs visualizing different anatomical structures using a few segmentations from one modality. This work is essential when we have limited numbers of segmented samples for one modality and potentially none for other modalities. To the best of our knowledge, this is the first work addressing this particular problem. Our results showed that our approach using registration-based data augmentation, unsupervised domain adaptation, and segmentation-based registration outperforms existing multimodal registration approaches. Future work will explore 3D extensions to our approach (see Sec. A for a related discussion on the limitations of our approach).

## Acknowledgments

This work was supported by NIH grants 1R01AR072013 and 1R01AR082684. The work expresses the views of the authors, not of the NIH. This research has been conducted using the UK Biobank Resource under Application Number 22783.

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

## Appendix A. Limitations

Our current approach has several limitations.

**2D Images.** Our 2D knee datasets are projections of 3D structures. This projection results in complex image intensity distributions because of the multi-layer structure of the knee, which cannot easily be disentangled based on a 2D registration (See Sec. C.4). This is the main reason why our approach currently uses segmentations for registration.

**Registration with Segmentation.** Although segmentations allow us to align object borders well, they disregard image texture. Hence, while our approach establishes correspondences inside the bones, those are purely based on a smooth interpolation of the deformation field.

**Training Cost.** Our approach involves several sequential training phases to train deep networks. This increases the training time.

**Future work.** As a next step, we will extend our work to 3D images. Unlike our 2D images, these are not projection images and will therefore allow us to incorporate a suitable loss function that can directly operate on images without solely relying on segmentations.

## Appendix B. Implementation and Network Parameters

### B.1. Dataset Splits

We crop left and right knee 2D X-ray images from the raw OAI dataset and select 123 random images from the dataset. Then, we resize both the DXA images from the UK Biobank and the OAI X-ray images to $256 \times 256$. Since segmentations are not provided for these datasets, we manually segment them into femur, tibia, and fibula. Additionally, we perform affine alignment on all images with respect to one randomly selected image, using the affine registration layers from (Greer et al., 2023). In unpaired image translation networks, the discriminator learns how anatomical regions should appear across a variety of images without seeing real paired examples. Affine alignment helps this process by standardizing the shapes and positions of these regions. Table 2 shows train/validation/test splits.

| Dataset | Train | Validation | Test |
|---------|-------|------------|------|
| DXA | $75 + 23^{\dagger}$ | 10 | 15 |
| X-ray | $75 + 23^{\dagger}$ | 10 | 15 |

Table 2: Dataset splits. † denotes samples without segmentations.

### B.2. Augmentation with Registration

We train a two-stage registration network (Tian et al., 2023) for each modality. We set the learning rate to $5 \times 10^{-5}$, use 20000 iterations per stage, use a batch size of 32, and penalize with localized normalized cross-correlation (LNCC) with $\sigma = 3$. For random elastic augmentation, we set the augmentation parameters to 3 control points and use a Gaussian with a standard deviation $\sigma$ of 3. We use a 2D Residual U-Net (Kerfoot et al., 2019) from

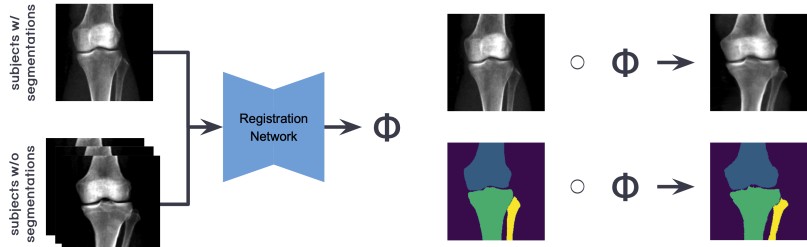

Figure 4: For augmentation, we first register images with segmentations $(I_m, S_m)$ to images without segmentations $I_f$. Then, we warp both $I_m$ and $S_m$ to create an artificial sample based on the predicted transformation map. *DXA images reproduced by kind permission of the UK Biobank®.*

the MONAI Framework (Cardoso et al., 2022) setting channels to (4, 8, 16, 32, 64, 128), strides to (1, 1, 1, 1, 1) and using 8 residual units.

## B.3. Domain Adaptation

We implement generators using ResNet (He et al., 2016), discriminators following the Patch-GAN (Isola et al., 2017) approach, and segmentation models using a Residual U-Net (Kerfoot et al., 2019) that shares the same architecture introduced in Section B.2. We embed our source segmentation network, which is pre-trained with our augmentation strategy, into the domain adaptation architecture and freeze it. Additionally, we initialize the segmentation network for the target modality with the same weights. We set $\lambda_{GAN}$ to 1, $\lambda_{cycle}$ to 10, and $\lambda_{sem}$ to 5 and train the network for 400 epochs with a batch size of 1. We use 1-Dice as a semantic loss $\mathcal{L}_{sem}$.

## B.4. Registration

For our registration task, we prefer using a single-stage registration network (Tian et al., 2023) over a two-stage network. Since registering segmentations is an easier task than registering images, even a single-stage model can converge easily and achieve high performance. We set the learning rate to $10^{-4}$, batch size to 32, and use 1-Dice as a similarity loss for 5000 steps. During inference, we do not employ any instance optimization.

## B.5. Computational Cost

We train our registration network for augmentation for ~4.5 hours, the segmentation network on the source dataset for ~1 hour, the UDA network for ~3.5 hours, and the registration network for registering segmentations for ~0.5 hours. We use one NVIDIA GeForce RTX 3090 for all of our experiments.

## Appendix C. Experimental Details

This section provides additional details and experimental results for the registration approaches 3-5 of Sec. 4. Specifically, Sec. C.1 provides experimental details for the multi-modal registration approach 3) where inputs are real images and our loss uses images, but only within manually segmented regions. Sec. C.2 provides experimental details for the multi-modal registration approach 4) where inputs are real images and our loss uses manual segmentations. Sec. C.3 provides experimental details for the multi-modal registration approach 5) where inputs are real-fake image pairs generated by our image translation network and the loss uses images. Sec. C.4 discusses the challenges of 2D knee registration and provides registration visualizations for the inner regions of the bones when using different similarity measures. Sec. C.5 provides experimental results for the applicability of our method to the case where we assume segmentations from both modalities are available. Sec. C.6 discusses the robustness of our domain adaptation strategy.

### C.1. Registration Training with Segmentation Constrained Similarity Loss

We design an experimental setting where the registration network takes real multimodal images as input, and the similarity loss is calculated using images but only within the segmented regions. First, we create binary masks from our manual segmentations by assigning 1 to segmented regions and 0 to the background. Then, we assume that image-mask pairs are given as moving $(I_m, M_m)$ and fixed images $(I_f, M_f)$. The registration network estimates the transformation map $\Phi = R(I_m, I_f; \theta) + id$ based on image intensities. If we denote the mapping from the moving image to the fixed image as $\Phi_\theta[I_m, I_f]$, the training loss function is

$$\ell = \sum_{i \in M_f} \mathcal{L}_{\text{sim}} \left( I_m \circ \Phi_\theta \left[ I_m, I_f \right], I_f \right)_i + \sum_{i \in M_m} \mathcal{L}_{\text{sim}} \left( I_f \circ \Phi_\theta \left[ I_f, I_m \right], I_m \right)_i + \qquad (5)$$
$$+ \lambda \left\| \nabla \left( \Phi_\theta \left[ I_m, I_f \right] \circ \Phi_\theta \left[ I_f, I_m \right] \right) - \mathbf{I} \right\|_F^2 ,$$

where $\mathcal{L}_{\text{sim}}()_i$ denotes the value of the similarity measure at location $i$. Here, we sum over all the foreground pixels of the fixed $(M_f)$ and moving $(M_m)$ image, respectively.

During training, we use gradient inverse consistency as the regularizer. Additionally, we conduct experiments with several similarity losses. During inference, this approach does not require any segmentations since it operates on image space.

Table 7 shows that our approach, with 3 segmentations from either the DXA or X-ray modality, outperforms this approach (registration approach 3 of Sec. 4) trained with several similarity losses while using 75 segmentations from each of the modalities. Therefore, optimizing over images even if constrained by segmentations, is not the best way for this registration task.

### C.2. Registration with Semantic Loss

Inspired by (Hu et al., 2018), we create an experiment where the registration network takes real multimodal images as input but where the similarity loss is based on manual segmentations, assuming manual segmentations are available for *all* images. Based on a

similar setting as in Sec. 3.4, we assume that image-segmentation pairs are given as moving $(I_m, S_m)$ and fixed images $(I_f, S_f)$. The registration network estimates the transformation map $\Phi = R(I_m, I_f; \theta) + id$ based on image intensities. If we denote the mapping from the moving image to the fixed image $\Phi_\theta[I_m, I_f]$, the training loss function is

$$\ell = \mathcal{L}_{\text{sim}}\left(S_m \circ \Phi_\theta\left[I_m, I_f\right], S_f\right) + \mathcal{L}_{\text{sim}}\left(S_f \circ \Phi_\theta\left[I_f, I_m\right], S_m\right) +$$
$$+ \lambda \left\|\nabla\left(\Phi_\theta\left[I_m, I_f\right] \circ \Phi_\theta\left[I_f, I_m\right]\right) - \mathbf{I}\right\|_F^2 . \tag{6}$$

We use gradient inverse consistency of the transformation maps as the regularizer and 1-Dice score on the segmentations as the similarity loss $\mathcal{L}_{\text{sim}}$. During inference, this approach does not require any segmentations since it operates on image space.

Table 7 shows that this approach (registration approach 4 of Sec. 4) performs worse when registering the fibula, even if we incorporate all manual segmentations for both modalities. We hypothesize that in contrast to a registration network that uses segmentations as inputs, a registration network that uses the acquired images directly will likely have to internally form some notion of a region of interest, essentially performing a form of latent segmentation. Hence, using the segmentations already as input is likely an easier task.

### C.3. Registration of Real-Fake Image Pairs

In our approach, we use a CycleGAN-based image translation network to train a segmentation network on the target domain, and we use this segmentation network for registration. However, we can also use our image translation network to convert our multimodal registration problem into a monomodal one. To obtain a strong competitor to our approach, we assume that we have manual segmentations for all images of both domains in contrast to our approach.

We train both a standard CycleGAN and a CycleGAN using manual segmentations for both modalities as semantic losses. Then, we translate all images in our training dataset to the same domain and create DXA & fake DXA and X-ray & fake X-ray image pairs. Based on these datasets, we train monomodal registration networks. During inference, we initially translate images to the same domain and then use these real-fake image pairs as the input to a monomodal registration network. Finally, we predict deformation fields.

Table 7 shows that registration via a CycleGAN using manual segmentations for both modalities as semantic losses is better than using a standard CycleGAN. However, because registering based on images is still difficult, the registration accuracy of our proposed approach remains better. Therefore, in our approach, it is beneficial to use the segmentation-guided CycleGAN to train the segmentation networks and work with the predicted segmentations.

Figure 5 illustrates the moving, fixed, fake fixed, and registered images produced using the CycleGAN monomodal registration approach with semantic loss for several cases. Table 7 presents the Dice scores of the CycleGAN monomodal registration approach, comparing results with and without the use of semantic loss.

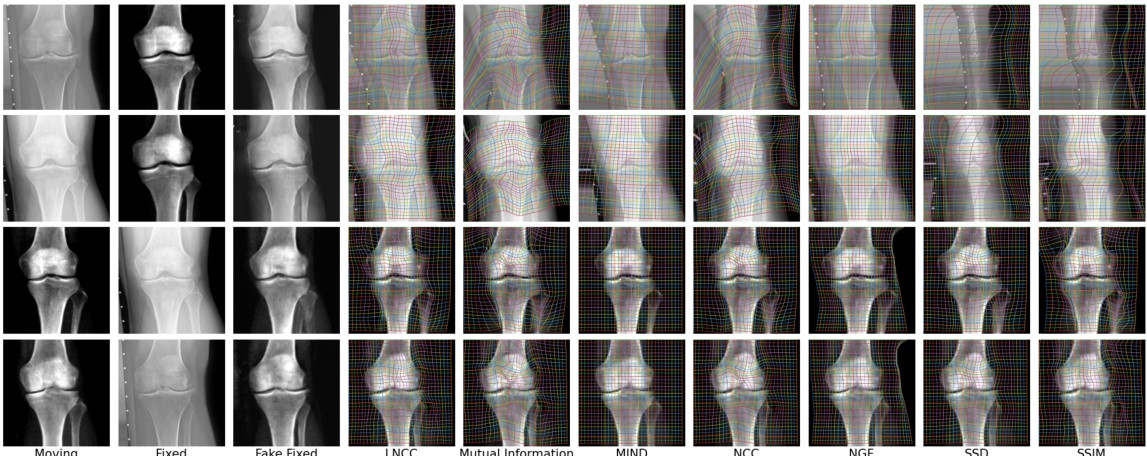

Figure 5: Qualitative results for multimodal registration with image translation. While registering between DXA and fake DXA images (bottom two rows) works reasonably well, registering between X-rays and fake X-rays (top two rows) is much harder and results in nonsensical registration results for some of the similarity measures. Best viewed zoomed. *DXA images reproduced by kind permission of the UK Biobank$^{®}$.*

## C.4. Registration of Anatomically Similar Pairs

In our registration setting, we register pairs based on their provided or predicted segmentations. Although registering based on segmentations aligns the borders of the bones well, it does not establish correspondences between the inner regions of the bones. To decouple registration problems due to visualization of different anatomical regions from registration problems due to image appearance differences for the different modalities within the same anatomical regions, we conducted an experiment by using images that only visualize common anatomical structures. We achieve this by masking the images of both modalities based on our manual segmentations. We thereby remove uncommon structures, in this case, soft tissue. This masking results in an artificial experimental dataset for multimodal registration that only shows anatomically similar regions. Then, we trained a registration network with several image similarity loss functions.

Figure 6 illustrates predictions of grid-based displacement fields based on these different image similarity measures. We observe each loss causes the network to behave significantly differently for inner bone regions due to appearance differences between the DXA and X-ray images. Combined with the multi-layer structure of the knee (including the presence of the patella), this leads to some similarity measures to highly spatially irregular deformation fields.

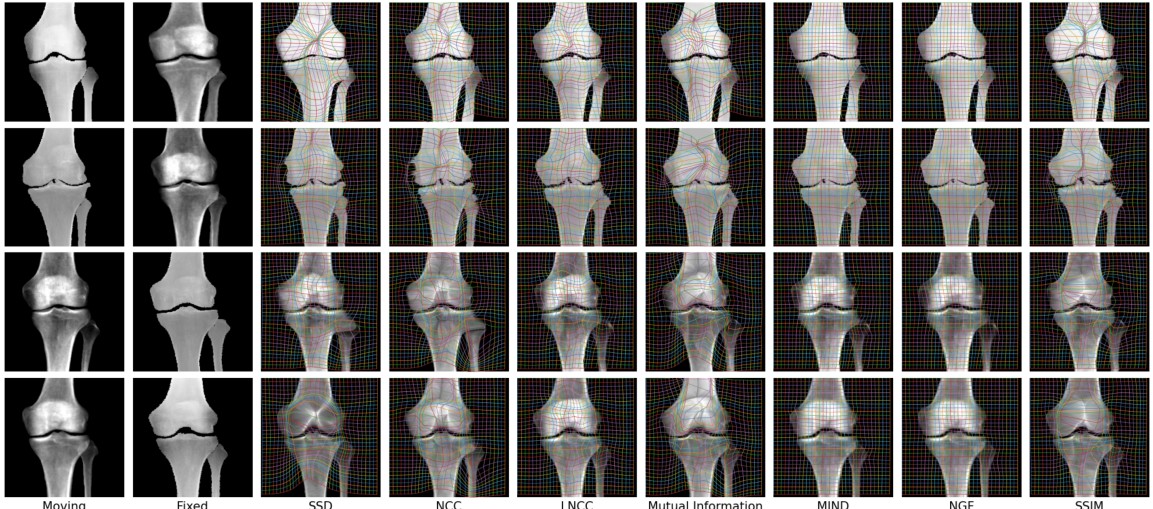

Figure 6: Qualitative results for multimodal registration using different similarity measures. The DXA and X-ray images have been masked to obtain pairs showing the same anatomical structure. Due to appearance differences within the bones, registration results strongly differ between the different similarity measures. *DXA images reproduced by kind permission of the UK Biobank®.*

## C.5. Registration where Segmentations for Both Modalities Exist

Our approach assumes that a small number of segmentations are provided for only one of the modalities. However, we also explore the applicability of our approach to a case where we have segmentations for both modalities. In this setting, the unsupervised domain adaptation phase is no longer necessary. Instead, we directly train segmentation networks for each modality with augmentation based on registration (see 3.2) for which we illustrated its effectiveness in Table 5. Then, we register image pairs again with predicted segmentations as we do in our proposed approach.

Table 3 compares two approaches: one that assumes segmentations are available from only one modality (ours) and another that utilizes segmentations from both modalities (ours w/o UDA). When the same number of segmentations are provided, our approach can outperform the approach without unsupervised domain adaptation (UDA) in some cases. For instance, ours reaches a 95.7% Dice score with 10 DXA segmentations and 96% with 10 X-ray segmentations, while ours w/o UDA with 5 segmentations for each modality only achieves a 94.2% Dice score. We observe similar results for the case of 20 segmentations: our approach reaches a 96.4% Dice score with 20 DXA segmentations and 96.5% with 20 X-ray segmentations while ours w/o UDA with 20 segmentations achieves a 96% Dice score. However, ours w/o UDA slightly outperforms with a 97% Dice score when 40 segmentations are provided, while ours reaches 96% with DXA and 96.8% with X-ray segmentations. These results demonstrate that our method, even when utilizing segmentations from only one modality, performs competitively compared to the approach

that benefits from segmentations from both modalities. This highlights the effectiveness of unsupervised domain adaptation in our approach.

| Approach | #DXA Seg. | #X-ray Seg. | DXA → X-ray | | | | | X-ray → DXA | | | | | Mean |
|---|---|---|---|---|---|---|---|---|---|---|---|---|---|
| | | | Femur | Tibia | Fibula | Mean | $\% \, |J_\varphi|$ | Femur | Tibia | Fibula | Mean | $\% \, |J_\varphi|$ | |
| Ours w/o UDA | 5 | 5 | 0.967 | 0.968 | 0.883 | 0.939 | 0.000 | 0.968 | 0.971 | 0.893 | 0.944 | 0.000 | 0.942 |
| | 10 | 10 | 0.971 | 0.969 | 0.941 | 0.960 | 0.000 | 0.973 | 0.970 | 0.935 | 0.959 | 0.000 | 0.960 |
| | 20 | 20 | 0.980 | 0.974 | 0.950 | 0.968 | 0.000 | 0.982 | 0.977 | 0.955 | 0.971 | 0.000 | 0.970 |
| Ours | 10 | ✗ | 0.972 | 0.972 | 0.925 | 0.956 | 0.000 | 0.972 | 0.973 | 0.928 | 0.958 | 0.000 | 0.957 |
| | 20 | ✗ | 0.976 | 0.970 | 0.944 | 0.963 | 0.000 | 0.976 | 0.970 | 0.945 | 0.964 | 0.000 | 0.964 |
| | 40 | ✗ | 0.971 | 0.967 | 0.941 | 0.960 | 0.000 | 0.972 | 0.968 | 0.941 | 0.960 | 0.000 | 0.960 |
| | ✗ | 10 | 0.976 | 0.967 | 0.937 | 0.960 | 0.000 | 0.977 | 0.969 | 0.935 | 0.960 | 0.000 | 0.960 |
| | ✗ | 20 | 0.976 | 0.970 | 0.945 | 0.964 | 0.000 | 0.977 | 0.973 | 0.950 | 0.967 | 0.000 | 0.965 |
| | ✗ | 40 | 0.979 | 0.971 | 0.952 | 0.967 | 0.000 | 0.980 | 0.973 | 0.954 | 0.969 | 0.000 | 0.968 |

Table 3: Class-based Dice scores for multimodal registration where we compare our approach (ours), which assumes manual segmentations from one of the modalities are provided for training, to ours w/o UDA, which assumes manual segmentations from both modalities are provided for training.

## C.6. Robustness of CycleGAN Based Unsupervised Domain Adaptation

Training a robust CycleGAN typically requires a large and diverse dataset. However, collecting such a dataset may be a challenging task. Under our experimental constraints, where we have a small number of images, this task becomes even more difficult. To investigate the robustness of our unsupervised domain adaptation approach, we run our experiments with 5 random seeds and report the mean and standard deviation of Dice scores for each class. Table 4 shows that our approach is reliable for domain adaptation, with the highest standard deviation being 0.028 for the worst-case scenario.

| #Segmentations | DXA → X-ray | | | X-ray → DXA | | |
|---|---|---|---|---|---|---|
| | Femur | Tibia | Fibula | Femur | Tibia | Fibula |
| 1 | $0.955 \pm 0.004$ | $0.935 \pm 0.007$ | $0.922 \pm 0.008$ | $0.984 \pm 0.002$ | $0.981 \pm 0.001$ | $0.922 \pm 0.028$ |
| 3 | $0.968 \pm 0.002$ | $0.962 \pm 0.002$ | $0.932 \pm 0.012$ | $0.986 \pm 0.001$ | $0.984 \pm 0.002$ | $0.967 \pm 0.001$ |
| 5 | $0.972 \pm 0.002$ | $0.969 \pm 0.002$ | $0.914 \pm 0.011$ | $0.986 \pm 0.003$ | $0.983 \pm 0.002$ | $0.969 \pm 0.001$ |
| 10 | $0.978 \pm 0.003$ | $0.976 \pm 0.004$ | $0.922 \pm 0.011$ | $0.988 \pm 0.001$ | $0.984 \pm 0.001$ | $0.964 \pm 0.002$ |
| 15 | $0.975 \pm 0.001$ | $0.970 \pm 0.004$ | $0.931 \pm 0.004$ | $0.987 \pm 0.001$ | $0.983 \pm 0.002$ | $0.966 \pm 0.004$ |
| 20 | $0.974 \pm 0.002$ | $0.970 \pm 0.002$ | $0.923 \pm 0.009$ | $0.986 \pm 0.002$ | $0.984 \pm 0.002$ | $0.966 \pm 0.004$ |
| 40 | $0.977 \pm 0.001$ | $0.967 \pm 0.004$ | $0.924 \pm 0.011$ | $0.988 \pm 0.001$ | $0.985 \pm 0.001$ | $0.965 \pm 0.001$ |
| 75 | $0.977 \pm 0.003$ | $0.971 \pm 0.004$ | $0.918 \pm 0.009$ | $0.985 \pm 0.001$ | $0.981 \pm 0.001$ | $0.963 \pm 0.006$ |

Table 4: Class-based Dice score means and standard deviations for unsupervised domain adaptation with 5 random training seeds. Dice scores are based on comparing the segmentation network results for the domain without available segmentations. The orange cell indicates a result discussed in the main text.

# Appendix D. Qualitative and Quantitative Results

This section provides the experimental results of our experiments. Specifically, Sec. D.1 provides a table comparing segmentation performances of networks without augmentation (raw), with elastic augmentation, and our augmentation with registration strategies. Sec. D.2 provides qualitative results that compare segmentation results without adaptation (w/o Adaptation) with several domain adaptation approaches: 1) "Baseline," which only uses images with available segmentations; 2) "Baseline + Predictions," which additionally uses pre-trained network predictions for all available images without segmentation; and 3) "Baseline + Augmentation," which augments the source dataset with our augmentation strategy. Additionally, it provides resulting visuals of our domain adaptation strategy ("Baseline+Augmentation") for a varying number of source dataset segmentations. Lastly, Sec. D.3 provides Dice scores and the percentage of voxels with a negative Jacobian ($\% |J_\varphi|$) for our registration approach compared to 1) an oracle approach; 2) multimodal registration where inputs are real images and our loss uses images; 3) multimodal registration where inputs are real images and our loss uses images but only within manually segmented regions; 4) multimodal registration where inputs are real images and our loss uses manual segmentations; 5) monomodal registration where inputs are real-fake image pairs generated by our image translation network, where loss uses images. Moreover, it provides registration visuals of our approach for a varying number of provided source dataset segmentations.

## D.1. Results of Augmentation Experiments

| | DXA | | | | | | | | | | | |
|---|---|---|---|---|---|---|---|---|---|---|---|---|
| | Raw | | | | Elastic | | | | Registration | | | |
| #Segmentations | Femur | Tibia | Fibula | Mean | Femur | Tibia | Fibula | Mean | Femur | Tibia | Fibula | Mean |
| 1 | 0.700 | 0.681 | 0.710 | 0.697 | 0.850 | 0.865 | 0.714 | 0.810 | 0.962 | 0.949 | 0.907 | **0.928** |
| 3 | 0.917 | 0.917 | 0.828 | 0.887 | 0.963 | 0.958 | 0.938 | 0.953 | 0.984 | 0.981 | 0.951 | **0.972** |
| 5 | 0.928 | 0.923 | 0.885 | 0.912 | 0.971 | 0.964 | 0.951 | 0.962 | 0.982 | 0.980 | 0.950 | **0.971** |
| 10 | 0.967 | 0.966 | 0.916 | 0.950 | 0.977 | 0.974 | 0.956 | 0.969 | 0.987 | 0.984 | 0.970 | **0.980** |
| 15 | 0.968 | 0.963 | 0.916 | 0.949 | 0.981 | 0.978 | 0.957 | 0.972 | 0.987 | 0.982 | 0.968 | **0.979** |
| 20 | 0.974 | 0.972 | 0.956 | 0.967 | 0.984 | 0.981 | 0.966 | 0.977 | 0.987 | 0.983 | 0.973 | **0.981** |
| 40 | 0.988 | 0.985 | 0.964 | 0.979 | 0.990 | 0.989 | 0.966 | 0.982 | 0.990 | 0.988 | 0.975 | **0.984** |
| 75 | 0.982 | 0.981 | 0.974 | **0.979** | 0.983 | 0.982 | 0.977 | 0.981 | 0.987 | 0.985 | 0.973 | **0.982** |
| | X-ray | | | | | | | | | | | |
| | Raw | | | | Elastic | | | | Registration | | | |
| #Segmentations | Femur | Tibia | Fibula | Mean | Femur | Tibia | Fibula | Mean | Femur | Tibia | Fibula | Mean |
| 1 | 0.736 | 0.607 | 0.243 | 0.529 | 0.729 | 0.706 | 0.345 | **0.593** | 0.772 | 0.666 | 0.291 | 0.576 |
| 3 | 0.763 | 0.745 | 0.421 | 0.643 | 0.882 | 0.878 | 0.643 | 0.801 | 0.941 | 0.936 | 0.811 | **0.896** |
| 5 | 0.778 | 0.751 | 0.642 | 0.724 | 0.900 | 0.882 | 0.699 | 0.827 | 0.946 | 0.920 | 0.759 | **0.875** |
| 10 | 0.911 | 0.900 | 0.736 | 0.849 | 0.945 | 0.941 | 0.885 | 0.924 | 0.976 | 0.959 | 0.918 | **0.951** |
| 15 | 0.953 | 0.941 | 0.837 | 0.910 | 0.980 | 0.970 | 0.887 | 0.946 | 0.976 | 0.967 | 0.948 | **0.964** |
| 20 | 0.956 | 0.956 | 0.879 | 0.930 | 0.982 | 0.978 | 0.932 | **0.964** | 0.978 | 0.975 | 0.935 | 0.963 |
| 40 | 0.976 | 0.964 | 0.902 | 0.947 | 0.977 | 0.979 | 0.939 | 0.965 | 0.982 | 0.980 | 0.955 | **0.972** |
| 75 | 0.977 | 0.981 | 0.934 | **0.964** | 0.982 | 0.986 | 0.956 | 0.975 | 0.984 | 0.985 | 0.959 | **0.976** |

Table 5: Class-based Dice scores for augmentation strategies evaluated on the DXA and X-ray segmentation tasks. Our method outperforms other strategies in most cases for a given number of available manual segmentations. The orange cells indicate the results discussed in the main text.

## D.2. Results of Domain Adaptation Experiments

| | DXA → X-ray | | | | | | | | | | | | | | | |
| | w/o Adaptation | | | | Baseline | | | | Baseline + Pred. | | | | Baseline + Aug. | | | |
| #Segmentations | Femur | Tibia | Fibula | Mean | Femur | Tibia | Fibula | Mean | Femur | Tibia | Fibula | Mean | Femur | Tibia | Fibula | Mean |
|---|---|---|---|---|---|---|---|---|---|---|---|---|---|---|---|---|
| Oracle | 0.984 | 0.985 | 0.959 | 0.976 | 0.984 | 0.985 | 0.959 | 0.976 | 0.984 | 0.985 | 0.959 | 0.976 | 0.984 | 0.985 | 0.959 | 0.976 |
| 1 | 0.689 | 0.635 | 0.232 | 0.519 | 0.944 | **0.954** | 0.742 | 0.880 | **0.953** | 0.938 | 0.899 | 0.930 | 0.952 | 0.932 | **0.910** | **0.931** |
| 3 | 0.657 | 0.672 | 0.272 | 0.534 | 0.969 | 0.963 | 0.856 | 0.929 | **0.976** | **0.969** | **0.937** | **0.961** | 0.968 | 0.965 | 0.903 | 0.945 |
| 5 | 0.611 | 0.388 | 0.331 | 0.443 | 0.971 | 0.969 | 0.897 | 0.946 | **0.972** | **0.972** | 0.889 | 0.944 | 0.967 | 0.966 | **0.911** | 0.948 |
| 10 | 0.788 | 0.759 | 0.358 | 0.635 | 0.978 | **0.978** | **0.928** | 0.961 | **0.979** | 0.974 | 0.897 | **0.950** | 0.977 | **0.978** | 0.891 | 0.949 |
| 15 | 0.782 | 0.729 | 0.476 | 0.662 | 0.973 | 0.965 | 0.926 | 0.955 | 0.975 | 0.970 | **0.936** | 0.960 | **0.977** | **0.971** | 0.934 | **0.961** |
| 20 | 0.729 | 0.780 | 0.533 | 0.681 | 0.977 | 0.972 | 0.926 | 0.958 | **0.981** | **0.975** | **0.932** | **0.963** | 0.980 | 0.973 | 0.920 | 0.958 |
| 40 | 0.807 | 0.837 | 0.562 | 0.735 | **0.978** | 0.965 | 0.932 | 0.958 | 0.975 | **0.969** | **0.940** | **0.961** | 0.977 | 0.968 | 0.920 | 0.955 |
| 75 | 0.698 | 0.615 | 0.462 | 0.592 | **0.979** | 0.974 | 0.909 | 0.954 | 0.978 | **0.978** | 0.929 | **0.962** | 0.972 | 0.976 | **0.930** | 0.959 |
| | X-ray → DXA | | | | | | | | | | | | | | | |
| | w/o Adaptation | | | | Baseline | | | | Baseline + Pred. | | | | Baseline + Aug. | | | |
| #Segmentations | Femur | Tibia | Fibula | Mean | Femur | Tibia | Fibula | Mean | Femur | Tibia | Fibula | Mean | Femur | Tibia | Fibula | Mean |
| Oracle | 0.987 | 0.985 | 0.973 | 0.982 | 0.987 | 0.985 | 0.973 | 0.982 | 0.987 | 0.985 | 0.973 | 0.982 | 0.987 | 0.985 | 0.973 | 0.982 |
| 1 | 0.696 | 0.720 | 0.438 | 0.618 | 0.971 | 0.964 | 0.855 | 0.930 | 0.976 | 0.878 | 0.064 | 0.639 | **0.982** | **0.978** | **0.930** | **0.954** |
| 3 | 0.820 | 0.871 | 0.843 | 0.845 | 0.982 | 0.981 | 0.954 | 0.972 | 0.981 | 0.978 | **0.970** | 0.976 | **0.987** | **0.983** | 0.964 | **0.978** |
| 5 | 0.929 | 0.947 | 0.845 | 0.907 | 0.985 | 0.980 | 0.958 | 0.974 | **0.988** | **0.982** | **0.969** | **0.980** | 0.986 | 0.981 | 0.967 | 0.978 |
| 10 | 0.936 | 0.967 | 0.886 | 0.930 | 0.985 | **0.985** | 0.953 | 0.974 | 0.985 | 0.980 | **0.973** | 0.979 | **0.990** | 0.984 | 0.970 | **0.981** |
| 15 | 0.886 | 0.923 | 0.818 | 0.876 | 0.985 | 0.984 | 0.968 | 0.979 | **0.990** | **0.985** | **0.976** | **0.984** | 0.989 | 0.984 | 0.963 | 0.979 |
| 20 | 0.927 | 0.925 | 0.928 | 0.927 | **0.988** | 0.984 | 0.970 | **0.981** | 0.985 | **0.987** | 0.972 | **0.981** | 0.987 | 0.985 | 0.969 | 0.980 |
| 40 | 0.957 | 0.958 | 0.944 | 0.953 | 0.988 | 0.983 | 0.966 | 0.979 | 0.988 | 0.984 | **0.972** | **0.981** | **0.989** | **0.985** | 0.970 | **0.981** |
| 75 | 0.940 | 0.958 | 0.781 | 0.893 | **0.988** | 0.980 | **0.972** | **0.980** | 0.986 | 0.981 | 0.971 | 0.979 | **0.988** | **0.982** | 0.967 | 0.979 |

Table 6: Class-based Dice scores for unsupervised domain adaptation for segmentation. We test our networks on the target modality with a limited number of source segmentations. Additionally, as an oracle method, we trained and tested a segmentation network on the target modality with full supervision. We observe that Dice scores converge when around 15 segmentations are provided. The results of domain adaptation are competitive with the oracle method. The orange cells indicate the results discussed in the main text.

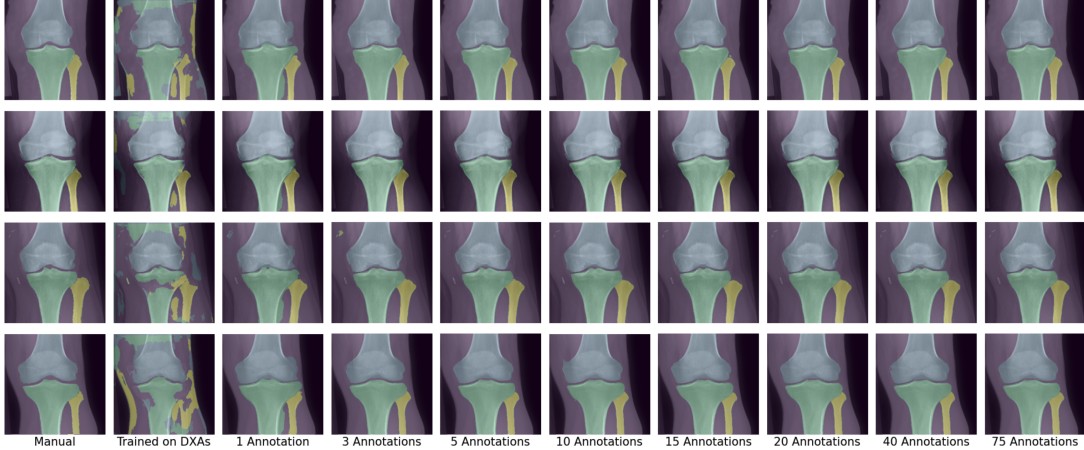

Figure 7: Qualitative results of our domain adaptation strategy ("Baseline+Augmentation") on the X-ray modality with varying numbers of provided DXA segmented samples.

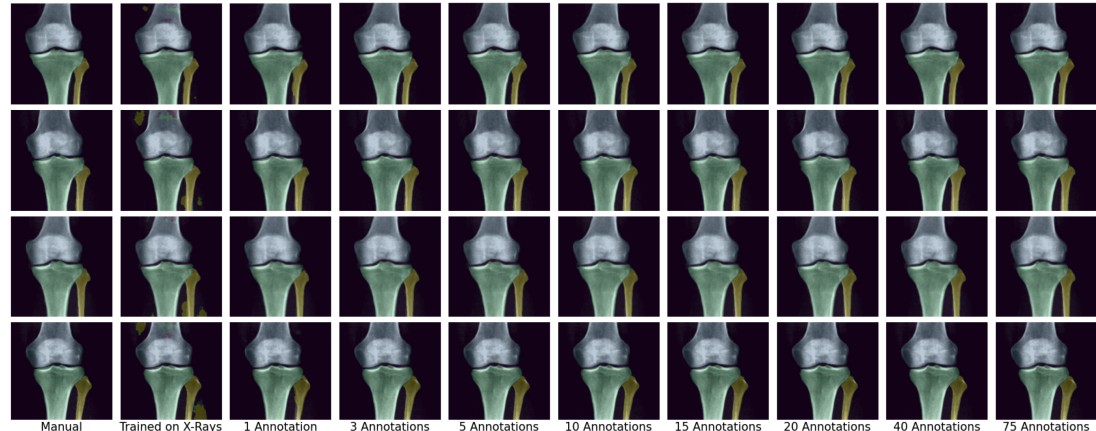

Figure 8: Qualitative results of our domain adaptation strategy ("Baseline+Augmentation") on the DXA modality for varying numbers of provided X-ray segmented samples. *DXA images reproduced by kind permission of the UK Biobank®.*

## D.3. Results of Registration Experiments

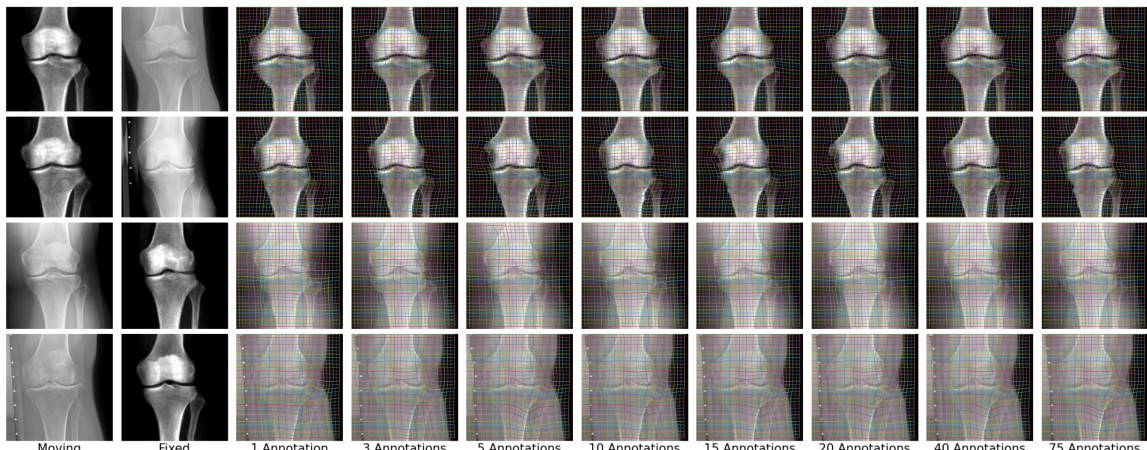

Figure 9: Qualitative results for our multimodal registration approach for different numbers of segmented DXA samples. *DXA images reproduced by kind permission of the UK Biobank®.*

| Approach | Input Type | #DXA Seg. | #X-ray Seg. | Loss | DXA → X-ray | | | | | X-ray → DXA | | | | | Mean |
|---|---|---|---|---|---|---|---|---|---|---|---|---|---|---|---|
| | | | | | Femur | Tibia | Fibula | Mean | %\|J_φ\| | Femur | Tibia | Fibula | Mean | %\|J_φ\| | |
| 1 | Manual Segmentations | 75 | 75 | Dice | 0.993 | 0.993 | 0.990 | 0.992 | 0.000 | 0.994 | 0.995 | 0.990 | 0.993 | 0.000 | 0.993 |
| 2 | Real DXA - Real X-ray | ✗ | ✗ | LNCC | 0.834 | 0.813 | 0.610 | 0.752 | 0.000 | 0.952 | 0.950 | 0.841 | 0.914 | 0.845 | 0.833 |
| | | ✗ | ✗ | MI | 0.658 | 0.770 | 0.460 | 0.629 | 0.030 | 0.787 | 0.948 | 0.634 | 0.790 | 7.422 | 0.710 |
| | | ✗ | ✗ | MIND | 0.866 | 0.893 | 0.528 | 0.762 | 0.000 | 0.847 | 0.888 | 0.493 | 0.867 | 0.000 | 0.753 |
| | | ✗ | ✗ | NCC | 0.643 | 0.721 | 0.399 | 0.588 | 0.000 | 0.727 | 0.907 | 0.357 | 0.664 | 7.308 | 0.626 |
| | | ✗ | ✗ | NGF | 0.678 | 0.712 | 0.368 | 0.586 | 0.000 | 0.758 | 0.794 | 0.356 | 0.636 | 0.000 | 0.611 |
| | | ✗ | ✗ | SSD | 0.671 | 0.656 | 0.177 | 0.501 | 0.000 | 0.698 | 0.664 | 0.164 | 0.509 | 5.309 | 0.505 |
| | | ✗ | ✗ | SSIM | 0.642 | 0.702 | 0.443 | 0.596 | 0.000 | 0.852 | 0.911 | 0.390 | 0.718 | 4.694 | 0.657 |
| 3 | Real DXA - Real X-ray | 75 | 75 | LNCC | 0.965 | 0.962 | 0.882 | 0.936 | 0.000 | 0.962 | 0.959 | 0.845 | 0.922 | 0.004 | 0.929 |
| | | 75 | 75 | MI | 0.964 | 0.962 | 0.874 | 0.933 | 0.000 | 0.963 | 0.960 | 0.856 | 0.926 | 0.001 | 0.930 |
| | | 75 | 75 | MIND | 0.698 | 0.445 | 0.310 | 0.484 | 0.000 | 0.735 | 0.645 | 0.285 | 0.555 | 1.597 | 0.520 |
| | | 75 | 75 | NCC | 0.964 | 0.963 | 0.875 | 0.934 | 0.001 | 0.964 | 0.961 | 0.848 | 0.924 | 0.002 | 0.929 |
| | | 75 | 75 | NGF | 0.966 | 0.963 | 0.869 | 0.933 | 0.000 | 0.964 | 0.960 | 0.865 | 0.930 | 0.000 | 0.931 |
| | | 75 | 75 | SSD | 0.597 | 0.563 | 0.000 | 0.387 | 99.737 | 0.630 | 0.769 | 0.000 | 0.466 | 75.812 | 0.427 |
| | | 75 | 75 | SSIM | 0.965 | 0.962 | 0.867 | 0.931 | 0.001 | 0.963 | 0.960 | 0.852 | 0.925 | 0.000 | 0.928 |
| 4 | Real DXA - Real X-ray | 75 | 75 | Dice | 0.966 | 0.965 | 0.863 | 0.931 | 0.000 | 0.966 | 0.967 | 0.844 | 0.926 | 0.000 | 0.929 |
| 5 | Real DXA - Fake DXA | ✗ | ✗ | LNCC | 0.970 | 0.964 | 0.802 | 0.912 | 0.000 | 0.964 | 0.960 | 0.817 | 0.914 | 0.000 | 0.913 |
| | | 75 | 75 | LNCC | 0.969 | 0.967 | 0.920 | 0.952 | 0.000 | 0.954 | 0.958 | 0.908 | 0.940 | 0.000 | 0.946 |
| | | ✗ | ✗ | MI | 0.969 | 0.963 | 0.770 | 0.901 | 0.001 | 0.966 | 0.960 | 0.799 | 0.908 | 0.002 | 0.905 |
| | | 75 | 75 | MI | 0.968 | 0.966 | 0.909 | 0.948 | 0.000 | 0.959 | 0.957 | 0.868 | 0.928 | 0.002 | 0.938 |
| | | ✗ | ✗ | MIND | 0.950 | 0.947 | 0.696 | 0.864 | 0.000 | 0.953 | 0.949 | 0.698 | 0.867 | 0.000 | 0.866 |
| | | 75 | 75 | MIND | 0.953 | 0.949 | 0.699 | 0.867 | 0.000 | 0.953 | 0.949 | 0.697 | 0.866 | 0.000 | 0.867 |
| | | ✗ | ✗ | NCC | 0.968 | 0.961 | 0.777 | 0.902 | 0.000 | 0.969 | 0.963 | 0.773 | 0.902 | 0.000 | 0.902 |
| | | 75 | 75 | NCC | 0.965 | 0.963 | 0.874 | 0.934 | 0.000 | 0.966 | 0.963 | 0.871 | 0.933 | 0.000 | 0.934 |
| | | ✗ | ✗ | NGF | 0.957 | 0.955 | 0.779 | 0.897 | 5.483 | 0.962 | 0.956 | 0.768 | 0.895 | 5.477 | 0.896 |
| | | 75 | 75 | NGF | 0.951 | 0.955 | 0.820 | 0.909 | 5.234 | 0.952 | 0.952 | 0.816 | 0.907 | 4.090 | 0.908 |
| | | ✗ | ✗ | SSD | 0.966 | 0.960 | 0.764 | 0.897 | 0.000 | 0.966 | 0.961 | 0.760 | 0.896 | 0.000 | 0.896 |
| | | 75 | 75 | SSD | 0.960 | 0.960 | 0.837 | 0.919 | 0.000 | 0.961 | 0.960 | 0.842 | 0.921 | 0.000 | 0.920 |
| | | ✗ | ✗ | SSIM | 0.968 | 0.964 | 0.775 | 0.902 | 0.000 | 0.961 | 0.958 | 0.791 | 0.903 | 0.000 | 0.903 |
| | | 75 | 75 | SSIM | 0.966 | 0.966 | 0.908 | 0.947 | 0.000 | 0.948 | 0.951 | 0.864 | 0.921 | 0.000 | 0.934 |
| | Real X-ray - Fake X-ray | ✗ | ✗ | LNCC | 0.966 | 0.962 | 0.806 | 0.911 | 0.000 | 0.966 | 0.960 | 0.818 | 0.915 | 0.000 | 0.913 |
| | | 75 | 75 | LNCC | 0.961 | 0.960 | 0.866 | 0.929 | 0.000 | 0.966 | 0.962 | 0.880 | 0.936 | 0.000 | 0.933 |
| | | ✗ | ✗ | MI | 0.947 | 0.950 | 0.761 | 0.886 | 0.000 | 0.952 | 0.949 | 0.796 | 0.899 | 0.000 | 0.893 |
| | | 75 | 75 | MI | 0.928 | 0.921 | 0.820 | 0.890 | 0.000 | 0.951 | 0.947 | 0.816 | 0.905 | 0.000 | 0.897 |
| | | ✗ | ✗ | MIND | 0.927 | 0.875 | 0.693 | 0.832 | 0.000 | 0.923 | 0.871 | 0.695 | 0.830 | 0.000 | 0.831 |
| | | 75 | 75 | MIND | 0.894 | 0.800 | 0.682 | 0.792 | 0.000 | 0.900 | 0.819 | 0.696 | 0.805 | 0.000 | 0.799 |
| | | ✗ | ✗ | NCC | 0.929 | 0.916 | 0.658 | 0.834 | 0.000 | 0.921 | 0.910 | 0.680 | 0.837 | 0.000 | 0.836 |
| | | 75 | 75 | NCC | 0.872 | 0.834 | 0.652 | 0.786 | 0.000 | 0.911 | 0.889 | 0.597 | 0.799 | 0.000 | 0.793 |
| | | ✗ | ✗ | NGF | 0.951 | 0.944 | 0.703 | 0.866 | 0.000 | 0.952 | 0.945 | 0.684 | 0.860 | 0.000 | 0.863 |
| | | 75 | 75 | NGF | 0.943 | 0.938 | 0.686 | 0.856 | 0.000 | 0.949 | 0.944 | 0.677 | 0.857 | 0.000 | 0.856 |
| | | ✗ | ✗ | SSD | 0.738 | 0.694 | 0.349 | 0.594 | 0.000 | 0.790 | 0.743 | 0.351 | 0.628 | 0.000 | 0.611 |
| | | 75 | 75 | SSD | 0.705 | 0.644 | 0.150 | 0.500 | 0.000 | 0.530 | 0.487 | 0.148 | 0.388 | 0.000 | 0.444 |
| | | ✗ | ✗ | SSIM | 0.878 | 0.836 | 0.562 | 0.759 | 0.000 | 0.901 | 0.864 | 0.571 | 0.779 | 0.000 | 0.769 |
| | | 75 | 75 | SSIM | 0.799 | 0.758 | 0.389 | 0.649 | 0.000 | 0.804 | 0.704 | 0.371 | 0.626 | 0.000 | 0.638 |
| Ours w/o UDA | Predicted Segmentations | 1 | 1 | Dice | 0.918 | 0.947 | 0.802 | 0.888 | 0.002 | 0.909 | 0.949 | 0.821 | 0.893 | 0.000 | 0.891 |
| | | 3 | 3 | Dice | 0.969 | 0.967 | 0.904 | 0.946 | 0.000 | 0.968 | 0.970 | 0.912 | 0.950 | 0.001 | 0.948 |
| | | 5 | 5 | Dice | 0.967 | 0.968 | 0.883 | 0.939 | 0.000 | 0.968 | 0.971 | 0.893 | 0.944 | 0.000 | 0.942 |
| | | 10 | 10 | Dice | 0.971 | 0.969 | 0.941 | 0.960 | 0.000 | 0.973 | 0.970 | 0.935 | 0.959 | 0.000 | 0.960 |
| | | 15 | 15 | Dice | 0.978 | 0.975 | 0.949 | 0.967 | 0.000 | 0.979 | 0.978 | 0.951 | 0.969 | 0.000 | 0.968 |
| | | 20 | 20 | Dice | 0.980 | 0.974 | 0.950 | 0.968 | 0.000 | 0.982 | 0.977 | 0.955 | 0.971 | 0.000 | 0.970 |
| | | 40 | 40 | Dice | 0.978 | 0.974 | 0.959 | 0.970 | 0.000 | 0.979 | 0.977 | 0.960 | 0.972 | 0.000 | 0.971 |
| | | 75 | 75 | Dice | 0.975 | 0.976 | 0.959 | 0.970 | 0.000 | 0.976 | 0.979 | 0.960 | 0.971 | 0.000 | 0.971 |
| Ours | Predicted Segmentations | 1 | ✗ | Dice | 0.956 | 0.945 | 0.899 | 0.933 | 0.000 | 0.957 | 0.944 | 0.902 | 0.934 | 0.000 | 0.934 |
| | | 3 | ✗ | Dice | 0.967 | 0.964 | 0.917 | 0.949 | 0.000 | 0.967 | 0.965 | 0.919 | 0.950 | 0.000 | 0.950 |
| | | 5 | ✗ | Dice | 0.970 | 0.966 | 0.932 | 0.956 | 0.000 | 0.971 | 0.967 | 0.934 | 0.957 | 0.000 | 0.957 |
| | | 10 | ✗ | Dice | 0.972 | 0.972 | 0.925 | 0.956 | 0.000 | 0.972 | 0.973 | 0.928 | 0.958 | 0.000 | 0.957 |
| | | 15 | ✗ | Dice | 0.975 | 0.971 | 0.938 | 0.961 | 0.000 | 0.976 | 0.972 | 0.939 | 0.962 | 0.000 | 0.962 |
| | | 20 | ✗ | Dice | 0.976 | 0.970 | 0.944 | 0.963 | 0.000 | 0.976 | 0.970 | 0.945 | 0.964 | 0.000 | 0.964 |
| | | 40 | ✗ | Dice | 0.971 | 0.967 | 0.941 | 0.960 | 0.000 | 0.972 | 0.968 | 0.941 | 0.960 | 0.000 | 0.960 |
| | | 75 | ✗ | Dice | 0.970 | 0.970 | 0.938 | 0.959 | 0.000 | 0.971 | 0.971 | 0.939 | 0.960 | 0.000 | 0.960 |
| | | ✗ | 1 | Dice | 0.917 | 0.942 | 0.793 | 0.884 | 0.002 | 0.906 | 0.946 | 0.825 | 0.892 | 0.008 | 0.888 |
| | | ✗ | 3 | Dice | 0.971 | 0.965 | 0.901 | 0.946 | 0.000 | 0.969 | 0.967 | 0.916 | 0.951 | 0.001 | 0.948 |
| | | ✗ | 5 | Dice | 0.966 | 0.965 | 0.889 | 0.940 | 0.000 | 0.966 | 0.965 | 0.905 | 0.945 | 0.000 | 0.943 |
| | | ✗ | 10 | Dice | 0.976 | 0.967 | 0.937 | 0.960 | 0.000 | 0.977 | 0.969 | 0.935 | 0.960 | 0.000 | 0.960 |
| | | ✗ | 15 | Dice | 0.976 | 0.971 | 0.941 | 0.963 | 0.000 | 0.977 | 0.974 | 0.945 | 0.965 | 0.000 | 0.964 |
| | | ✗ | 20 | Dice | 0.976 | 0.970 | 0.945 | 0.964 | 0.000 | 0.977 | 0.973 | 0.950 | 0.967 | 0.000 | 0.965 |
| | | ✗ | 40 | Dice | 0.979 | 0.971 | 0.952 | 0.967 | 0.000 | 0.980 | 0.973 | 0.954 | 0.969 | 0.000 | 0.968 |
| | | ✗ | 75 | Dice | 0.977 | 0.972 | 0.950 | 0.966 | 0.000 | 0.978 | 0.975 | 0.953 | 0.969 | 0.000 | 0.968 |

Table 7: Class-based Dice scores for multimodal registration. This table shows quantitative results for our approach comparing the following approaches: 1) An oracle approach (to provide an upper performance bound); 2) Multimodal registration where inputs are real images and our loss uses images; 3) Multimodal registration where inputs are real images and our loss uses images but only within manually segmented regions; 4) Multimodal registration where inputs are real images and our loss uses manual segmentations; 5) Monomodal registration where inputs are real-fake image pairs generated by our image translation network, where loss uses images.

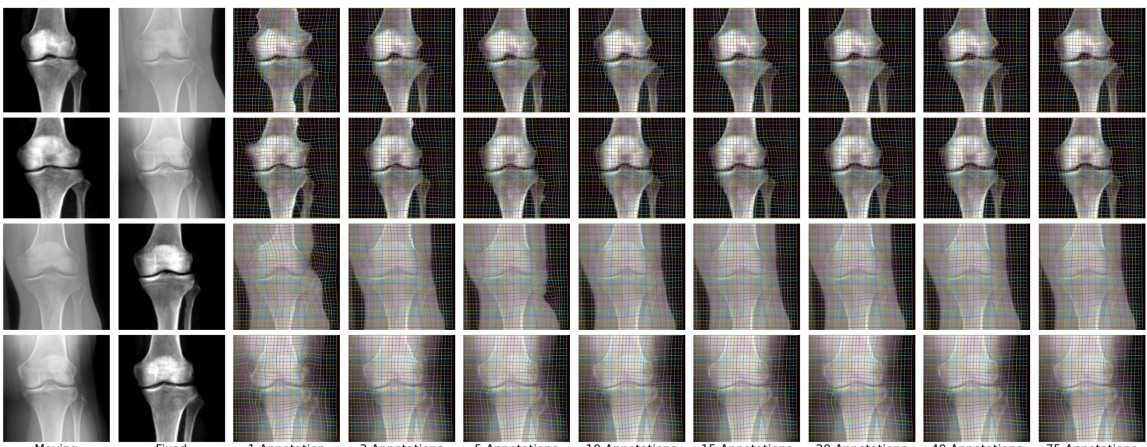

Figure 10: Qualitative results for our multimodal registration approach for different numbers of segmented X-ray samples. *DXA images reproduced by kind permission of the UK Biobank®.*

