# OpenReview forum: "Multimodal Image Registration Guided by Few Segmentations from One Modality"
_MIDL.io/2024/Conference — MIDL 2024 Poster_

### Official Review · Reviewer_qPPp · 2024-02-26

**Confidence:** 5
**Preliminary Rating:** 3
**Recommendation:** Poster
**Final Rating:** 4

**Summary:**

This paper introduces a novel pipeline for multimodal image registration, particularly effective when limited segmentation data is available for one modality. The authors hypothesize that a registration model, when trained with supervision from segmentations across both modalities, can enhance focus on common image regions, thereby improving registration accuracy.
Proposed Pipeline Overview:
1.	Augmenting modality/dataset A with segmentations using a registration network
2.	Train segmentation network on dataset A
3.	Train a segmentation network on modality B using guidance an unsupervised domain adaptation network
4.	Train registration network which registers segmentations of modality A and B

The authors apply their pipeline on the task of registering 2D knee DXA and X-ray image pairs showing promsing results on this dataset. However, the experimental setting is not completely clear.

**Strengths:**

-	The paper addresses a unique and challenging scenario in image registration, offering a comprehensive strategy to tackle the issue when only one modality has available segmentation.
-	The paper is well-motivated, with a hypothesis that is clearly articulated and easy to comprehend.
-	Preliminary results indicate the pipeline's potential to serve as a versatile solution across various applications.

**Weaknesses:**

-	The clarity of the experimental design is lacking, with a disconnect between the experiments listed in Table 1 and their detailed descriptions, complicating the understanding of the methodologies employed (see detailed comments)
-	For the presented registration application of 2D knee DXA and X-ray images, no information is provided why those images need to be registered.
-	The necessity for the proposed pipeline, specifically for the chosen modalities, is not adequately justified. The use case appears somewhat contrived, with an unclear rationale behind the specific choice of segmentation and training approach, raising questions about the necessity of the unsupervised domain adaptation.
-	The discussion on the pipeline's limitations is notably brief and could benefit from a more detailed exploration.

**Detailed Comments:**

•	The experiment section's structure could be improved for better clarity, especially in how the different experimental setups relate to the proposed method and previously established methods. The authors describe 4 different experimental settings in the Experiment section:  “1) Oracle model that utilizes manual segmentations for each pair; 2) Multimodal registration where inputs are real images and a penalty term based on intensities; 3) Monomodal registration where inputs are realfake image pairs generated by an image translation network, with a penalty term based on intensities (See Sec. B.2); 4) Multimodal registration where inputs are real images and penalty terms based solely on segmentation masks”. Additionally, they say that “we train a registration network (Tian et al., 2023) capable of aligning segmentations.”. Are 1) to 4) all using the same method described in (Tian et al., 2023)?  In Table 1, list two types of approaches, either GradICON + similarity measure or ‘ours’. Are all GradICON referring to (Tian et al., 2023)? If so add this in the description! What does ‘ours’ referring to? Is it actually a new registration method? If so, which regularizer was then used? For 3), there are multiple rows given in the table, e.g. Real “DXA - Fake DXA 75 75 GradICON + SSIM “ – however, it is not completely clear for what the 75 segmentation masks has been used. Does 4) correspond to ‘ours’ in the table? If so, it doesn’t fit because in 4) the images are used as input and the network is guided by the segmentation masks in the penalty, whereas ‘ours’ uses the ‘predicted segmentations’ as input. Please clarify and rework the experiment section to make it easier to follow.
•	“Manual segmentations are used as an oracle model for registration.” – Why is this experiment only performed with the baseline registration method and not with the suggested method or is the actual registration method the same (see above)? Using the manual segmentation in the proposed pipeline would give an upper bound on the registration performance. In my opinion that is a very valuable experiment to add. If this is exactly what the authors did, please clarify!
•	The paper could explore additional experiments where similarity measures are constrained by available structures in one modality, which could further validate the pipeline's efficacy.
•	A discussion on the practicality of manual segmentation in real-world scenarios would enhance the paper's relevance, alongside a clearer argument for when unsupervised domain adaptation is essential: “Moreover, manual segmentations are typically available for only a limited number of images and modalities”, please give examples where this is the case and why it is not an option to annotate on each modality a few samples and only use your registration augmentation. This could be an interesting point for the discussion. To me, UDA is not necessary in this example. In which cases it is necessary?
•	Revisiting the hypothesis in the discussion to explicitly address its validation or rejection based on the experimental results would strengthen the paper's narrative.

**Justification Of Final Rating:**

The authors provided a detailed and clear explanations during the rebuttal. Integrating this information into the original manuscript will undoubtedly enhance its quality and improve its comprehensibility.  On the condition that the explanations and improvements are incorporated into the original manuscript, I recommend that the paper be accepted.

**Justification Of The Preliminary Rating:**

The proposed pipeline is interesting and the paper has potential. However, in the current state, it is hard to follow and the added value is not clear. If the authors to manage those during the rebuttal period, the paper could be of interested for the MIDL community.

**Questions To Address In The Rebuttal:**

-	Please rework the experiment section to make it easier to follow.
-	Please add the experiment restricting the similarity measure to the available structures on the fixed image. This experiment should be early implemented and don’t take too much time.
-	Please address as many points as possible from the detailed comments.
-	Add a more intensive discussion on limitations of the work.

**Special Issue:**

No

---

> ### Author Response · Authors · 2024-03-18
> **Answer to Reviewer qPPp**
>
> We thank the reviewer for the time, advice, and feedback. We address the main questions and issues below:
>
> **-The clarity of the experimental design is lacking, with a disconnect between the experiments listed in Table 1.**
>
> We agree, and thank you for pointing out the clarity issue. We have edited the methodology section to enhance understanding and readability. Also, we added numbers to the tables to make it easier to match text with results. Here is a summary of our baseline approaches and the questions that we asked ourselves while designing these approaches:
>
> 1. **An oracle approach that has access to manual segmentations for all images and uses manual segmentations as input and for the loss:** We assume all manual segmentations are provided and perform registration based on these segmentations. This experiment tests the ideal scenario with correct segmentations on both modalities, aiming to achieve an upper bound of registration accuracy achievable with high-quality segmentations. **Question:** What level of registration accuracy can be achieved using manual segmentations for registration?
>
> 2. **Multimodal registration where inputs are real images and our loss uses images:** This baseline approach tests common registration practices where the network attempts registration based on a penalty term applied to intensity values. We tested various registration losses to observe their behavior when image pairs depict different anatomical structures. **Question:** What level of registration accuracy can be achieved using common registration practices that work directly with images?
>
> 3. **Multimodal registration where inputs are real images and our loss uses images but only within manually segmented regions:** Thanks to your valuable suggestion, we now also conducted additional experiments by constraining similarity measures to available structures in one modality. Here, we optimize our network by calculating the loss only over segmented regions of the images. This is a form of loss function masking. **Question:** What level of registration accuracy can be achieved if one knew exactly where to look for the structure of interest?
>
> 4. **Multimodal registration where inputs are real images and our loss uses manual segmentations:** We investigated using images as inputs and optimizing the network with their corresponding segmentations, testing the necessity of using segmentations as inputs. **Question:** What level of registration accuracy can be achieved by penalizing the network with segmentations while providing real images as inputs?
>
> 5. **Monomodal registration where inputs are real-fake image pairs generated by our image translation network, where loss uses images:** Here, we convert our multimodal problem into a monomodal setting using cycleGAN and an additional semantic loss. We generate image pairs consisting of a real image and a fake image translated to the modality of the real image, exploring the registration of artificially created monomodal image pairs. **Question:** What level of registration accuracy can be achieved by converting our multimodal image registration task to a monomodal setting with image translation?
>
> 6. **Ours:** It refers to our full proposed approach. We first augment the source dataset, then train a segmentation network on the source dataset, then perform unsupervised domain adaptation to obtain a segmentation network for the target dataset, and finally register images based on predictions of the segmentation networks.
>
> **-For the presented registration application of 2D knee DXA and X-ray images, no information is provided why those images need to be registered.**
>
> Our goal is to support large-scale imaging genetics studies. To obtain large sample sizes, pooling different studies is useful. However, these different studies may not use the same kind of imaging. For example, the Osteoarthritis Initiative provides knee radiographs whereas the UK Biobank only has DXA images. By being able to register these images (ultimately to a common atlas space) we will be able to extract image biomarkers that are *spatially corresponding* which we will then be able to relate to gene expression. We will add this motivation to the final paper.
>
> **-The necessity of the unsupervised domain adaptation.**
>
> Since we are training based on segmentations and we do not have segmentations for one of the modalities, UDA is necessary. Please see the answer above as to why we are targeting this particular registration problem.

---

> > ### Author Response · Authors · 2024-03-18
> > **Answer to Reviewer qPPp - continued**
> >
> > **- The discussion on the pipeline's limitations is notably brief and could benefit from a more detailed exploration.**
> >
> > Limitations are explained in the Appendix A. We are now also discussing computational cost in the paper. The main limitation of our approach is that it only focuses on bone shape (via the segmentations) and does not attempt to register the bone interiors based on image information. The primary reason for this limitation is the multi-layered structure of the knees. Since we are working with 2D images that are projected versions of 3D volumes, they are insufficient to accurately capture anatomical structures. Therefore, we have chosen to register only the bone shapes using predicted segmentation masks. Additionally, our approach consists of several phases, and the success of the registration depends on the performance of each trained intermediate model. Although we have demonstrated the robustness of unsupervised domain adaptation (UDA) in our case (see Appendix C.6), UDA may perform poorly in other more complex domains, which may in turn lead to inaccurate registrations or even registration failures.
> >
> > **- Confusion about the choice of the regularizer and the architecture across the experiments and what ours refers to?**
> >
> > We are sorry about the confusion. Yes, all GradICON results refer to the approach by (Tian et al., 2023) and we use the same architecture and regularization for all of our experiments. We only change input types and how we measure the similarity. Ours indicates using the GradICON registration model with segmentations as input which are obtained via our unsupervised domain adaptation approach. To prevent any future confusion, we have redesigned Table 1 and Table 7.
> >
> > **- For 3), there are multiple rows given in the table, e.g. Real “DXA - Fake DXA 75 75 GradICON + SSIM “ – however, it is not completely clear what the 75 segmentation masks have been used for.**
> >
> > For our approach, we use the CycleGAN to obtain segmentations which we then use for registration. Here, manual segmentations are only provided for one domain and then used in the cycleGAN to provide a semantic loss to train segmentation networks for **both** modalities. In the Real DXA - Fake DXA experiments we explore the registration quality when converting the multi-modal registration problem to a monomodal registration problem by image translation via a CycleGAN. To obtain the strongest comparable competitor to our approach we assume here that we have manual segmentations for **both** domains (which is precisely what our approach tries to avoid). We see that registration via a CycleGAN using manual segmentations for both modalities as semantic losses is indeed better than using a standard CycleGAN (without using the manual segmentations during training). However, because registering based on images is still difficult, the registration accuracy of our proposed approach remains better. In effect, we show that for our setting it is beneficial to use the segmentation-guided CycleGAN to train the segmentation networks and then work with these segmentations; the synthesized images are only a byproduct for us that we do not end up using. We also discussed this approach in detail in the **Appendix C.3**.
> >
> > **- Does 4) correspond to ‘ours’ in the table? If so, it doesn’t fit because in 4) the images are used as input and the network is guided by the segmentation masks in the penalty, whereas ‘ours’ uses the ‘predicted segmentations’ as input.**
> >
> > Thanks for pointing out this potential source of confusion. Ours refers to our proposed approach and it does not correspond to 4. We reworked the experiment section and tables to make the flow more clear.
> >
> > **- “Manual segmentations are used as an oracle model for registration.” – Why is this experiment only performed with the baseline registration method and not with the suggested method or is the actual registration method the same (see above)? Using the manual segmentation in the proposed pipeline would give an upper bound on the registration performance.**
> >
> > All baseline approaches and our approach use the same registration network and regularizer. When segmentations are already available (i.e., the setup of the Oracle experiment), we do not require the augmentation and UDA phases which are meant to train a segmentation network. The Oracle model uses gradICON in the same way as all the other approaches we compare to. The only difference is that we assume manual segmentations are available during the inference. I.e., segmentations for registrations are not provided by a trained segmentation network.

---

> > > ### Author Response · Authors · 2024-03-18
> > > **Answer to Reviewer qPPp - continued**
> > >
> > > **- Please add the experiment restricting the similarity measure to the available structures on the fixed image.**
> > >
> > > Thank you for suggesting an additional experimental setting, we added this scenario to our experiments, please see Appendix C.1 for more details. Table 7 shows that our approach, with 3 segmentations from either the DXA or X-ray modality, outperforms this approach (registration approach 3 of Sec. 4) trained with several similarity losses while using 75 segmentations from each of the modalities. Therefore, optimizing over images, even if constrained by segmentations, is not the best way for this registration task.
> > >
> > > **- A discussion on the practicality of manual segmentation in real-world scenarios would enhance the paper's relevance. Please give examples where this is the case and why it is not an option to annotate on each modality a few samples and only use your registration augmentation.**
> > >
> > > Our approach is motivated by registering many different datasets. Where images can come from different modalities, may have been acquired on different scanners, or with different imaging protocols. With the ultimate goal of supporting the pooled analysis of several large-scale studies (such as data from the UK Biobank, the Osteoarthritis Initiative, the Multicenter Osteoarthritis Study (MOST), and the Johnston County Study for example). All these studies are longitudinal and during such studies (e.g., for MOST) imaging protocols may also change over time as new, improved imaging capabilities become available. While it is true that one could segment all relevant structures on all different kinds of images, our goal was to develop an approach that could ultimately be used much more flexibly in such a scenario without manual (2D and 3D) segmentations for each image type.
> > >
> > > **- To me, UDA is not necessary in this example. In which cases it is necessary? Revisiting the hypothesis in the discussion to explicitly address its validation or rejection based on the experimental results would strengthen the paper's narrative.**
> > >
> > > Taking into account your valuable feedback, we now also conducted an ablation study where we assumed that a small number of segmentations are provided for both modalities. As you suggested, in this setup, we no longer need unsupervised domain adaptation (UDA), and our augmentation strategy is sufficient for training segmentation networks for both domains. In Appendix C.5, we reported the registration performance of our strategy without the UDA phase.
> > >
> > > We observe that our approach can outperform our approach without unsupervised domain adaptation (UDA) when the same number of segmentations are provided in some cases. For instance, ours reaches a 95.7\% Dice score with 10 DXA segmentations and 96\% with 10 X-Ray segmentations, while ours w/o UDA with 5 segmentations for each modality only achieves a 94.2\% Dice score. We observe similar results for the case of 20 segmentations as well. However, ours w/o UDA slightly outperforms with a 97\% Dice score when 40 segmentations are provided, while ours reaches 96\% with DXA and 96.8\% with X-ray segmentations. These results demonstrate that our method, even with segmentation from only one modality, shows competitive performance compared to the approach that benefits from segmentations from both modalities and highlights the effectiveness of unsupervised domain adaptation.

---

> > > > ### Comment · Reviewer_qPPp · 2024-03-21
> > > >
> > > > Thank you very much for providing such detailed and clear explanations. Integrating this information into the original manuscript will undoubtedly enhance its quality and improve its comprehensibility. Additionally, the rebuttal has effectively addressed all of my outstanding questions. Consequently, I am adjusting my evaluation to a weak accept.

---

> > > > > ### Author Response · Authors · 2024-03-26
> > > > > **Thanks to Reviewer qPPp**
> > > > >
> > > > > Thank you for your detailed reviews, feedback, and raising the score. Your insightful suggestions have helped us improve the clarity and quality of our work. We truly appreciate your time and patience in reviewing our paper.

---

### Official Review · Reviewer_J4kr · 2024-02-28

**Confidence:** 4
**Preliminary Rating:** 4
**Recommendation:** Poster
**Final Rating:** 4

**Summary:**

This paper tackles the complex issue of unsupervised multimodal image registration with an innovative approach that leverages a small set of labels available for one modality. The proposed method involves a three-step pipeline. Initially, a segmentation network is trained on the modality with labels, utilizing an augmented training and label the entire dataset. Subsequently, an image-to-image translation network is trained to convert images between the two modalities, incorporating an image segmentation objective to maintain anatomical accuracy. This network facilitates the generation of labels for the second modality. The final step involves training an unsupervised image registration network using these generated segmentation masks, making the registration process modality-independent. The methodology is validated on knee images from two distinct modalities.

**Strengths:**

The challenge of defining a similarity metric for unsupervised multimodal image registration is significant, and the paper presents an innovative solution. By employing a combination of deep learning techniques within a structured pipeline, this approach addresses the challenge in a novel way.

**Weaknesses:**

The paper could be strengthened by including experiments on a broader variety of images, such as 3D brain scans, to demonstrate the method's applicability across different anatomical regions. Additionally, the rationale behind using only labels for registration, rather than directly inputting the original images for a more authentic multimodal registration, remains unclear to me.

**Detailed Comments:**

Refer to "Weaknesses" for suggestions on areas requiring further clarification or expansion. Specifically, a detailed justification for the methodological choice of not using the images as inputs for registration would enhance the reader's understanding of the approach's advantages.

**Justification Of Final Rating:**

Having read the updated manuscript, I have no further questions at this time. I appreciate the availability of the source code. However, the paper lacks sufficient novelty to justify a 'strong accept,' so I will remain with my previous decision of 'weak accept.' Great work!

**Justification Of The Preliminary Rating:**

The paper addresses a significant challenge within the field and proposes a well-articulated and intriguing solution. However, the impact and generalizability of the findings could be further substantiated with additional experiments across varied datasets, such as 3D brain imaging studies.

**Questions To Address In The Rebuttal:**

Could you clarify if there was any experimentation using the original images as inputs for the registration network, and the labels only for optimization? This clarification might address potential confusion regarding the chosen methodology.

**Special Issue:**

No

---

> ### Author Response · Authors · 2024-03-18
> **Answer to Reviewer J4kr**
>
> We thank the reviewer for the time, advice, and feedback. We address the main questions and issues below:
>
> **-The paper could be strengthened by including experiments on a broader variety of images, such as 3D brain scans.**
>
> We agree. This is a first step and we are planning to extend our approach to 3D in future work.
>
> **-What is the rationale behind using only labels for registration, rather than directly inputting the original images for a more authentic multimodal registration?**
>
> The rationale for using segmentations only is based on the specific DXA/radiograph registration task. As these are projection images we are dealing with layered anatomical structures. Hence, we opted for registering the bone shapes only. However, our approach is general and could also be used with images instead of segmentations as inputs. This is what we are planning in future work for 3D registration where projection issues are no longer present.
>
> **-Could you clarify if there was any experimentation using the original images as inputs for the registration network, and the labels only for optimization?**
>
> Note that Table 1 (Real DXA - Real X-ray - 75 75 - Dice) refers to the case where we pass real images to the network and penalize them based on their segmentations. This scenario is also mentioned as “*4) Multimodal registration where inputs are real images and our loss uses manual segmentations*” in the experiments section. Additionally, we have now provided a more detailed explanation in **Appendix C.2**. In this scenario, we observe that the registration network performs significantly worse when registering the fibula, even if we incorporate all manual segmentations for both modalities. Our hypothesis is that in contrast to a registration network that uses segmentations as inputs, a registration network that uses the acquired images directly will likely have to internally form some notion of a region of interest, essentially performing a form of latent segmentation. Hence, using the segmentations already as input is likely an easier task.

---

> > ### Author Response · Authors · 2024-03-26
> > **Kind Reminder to Reviewer J4kr**
> >
> > Thank you again for sharing your valuable feedback and comments with us. As the discussion period ends soon, we kindly request that you let us know if our rebuttal and the updated manuscript have addressed your concerns. We would love to clarify and explain any remaining concerns you may have.

---

> > > ### Comment · Reviewer_J4kr · 2024-03-27
> > >
> > > Thank you very much for revising the manuscript and addressing my questions. Having read the updated manuscript, I have no further questions at this time. I also appreciate the availability of the source code.

---

> > > > ### Author Response · Authors · 2024-03-27
> > > > **Thanks to Reviewer J4kr**
> > > >
> > > > Thank you for your reviews and kind feedback! Your suggestions have helped us improve the clarity of our work. We appreciate your time in reviewing our paper.

---

### Official Review · Reviewer_vdWQ · 2024-03-02

**Confidence:** 4
**Preliminary Rating:** 4
**Recommendation:** Poster

**Summary:**

The authors present a novel registration and segmentation approach tailored to the challenging scenario of dealing with different anatomical structures and the limited availability of segmentations. The authors assume that only a small number of labeled examples is available for the source domain, and no examples for the target domain. The proposed algorithm proceeds in three stages: First, the small number of labeled examples for the source domain is augmented using unsupervised deep registration. Second, a domain adaptation approach by Gogoll et all. is applied to train a segmentation algorithm on the target domain. Third, an inverse-consistent registration network by Tian et al. is used to register the segmentations for the source and target domain.

**Strengths:**

The paper is well-written and easy to understand. The suggested approach seems novel and addresses the relevant problem of registering image pairs from different modalities that each highlight different anatomical structures. The appendix offers extensive ablation studies to provide more insights into the design decisions.

**Weaknesses:**

The paper leverages and combines multiple existing techniques as described and cited in the manuscript (Grogoll et al. and Tian et al.'s work in particular). This may reduce the novelty of the overall approach.

**Detailed Comments:**

No further comments.

**Justification Of The Preliminary Rating:**

The authors propose a novel registration and segmentation approach tailored to the challenging scenario of the limited availability of segmentations for different anatomical structures. The approach utilizes unsupervised deep registration, domain adaptation, and inverse-consistent registration to register image pairs from different modalities. Despite leveraging existing techniques, the authors provide extensive ablation studies in the appendix, demonstrating the effectiveness of their proposed approach.

**Questions To Address In The Rebuttal:**

Could you provide more details on the robustness of training the CycleGAN?
Will the code be made publicly available upon publication?
Can you give more details on the computational cost of training this method?

---

> ### Author Response · Authors · 2024-03-18
> **Answer to Reviewer vdWQ**
>
> We thank the reviewer for the time, advice, and feedback. We address the main questions and issues below:
>
> **-Details on the robustness of training the CycleGAN.**
>
> We have conducted additional experiments to demonstrate the robustness of our domain adaptation network by training it with **5 different seeds**. The mean and standard deviation of Dice scores are now reported in **Appendix C.6**. Based on these results, our approach is reliable for domain adaptation, with the highest standard deviation being **0.028 in the worst-case scenario** (where only 1 manual segmentation is provided). With the incorporation of 15 segmentations, the standard deviation is as low as **0.004**.
>
> **-Will the code be made publicly available upon publication?**
>
> Yes, it will be publicly available at: https://github.com/uncbiag/SegGuidedMMReg. We also added this link to the abstract.
>
> **-Details on computational cost of training this method.**
>
> We train our registration network for augmentation for $\sim$4.5 hours, the segmentation network on source dataset for $\sim$1 hour, the UDA network for $\sim$3.5 hours, and the registration network for registering segmentations for $\sim$0.5 hours. We use one NVIDIA GeForce RTX 3090 for all of our experiments. We also added this information to the paper in Appendix B.5.

---

> > ### Author Response · Authors · 2024-03-26
> > **Kind Reminder to Reviewer vdWQ**
> >
> > Thank you again for sharing your valuable feedback and comments with us. As the discussion period ends soon, we kindly request that you let us know if our rebuttal and the updated manuscript have addressed your concerns. We would love to clarify and explain any remaining concerns you may have.

---

> > > ### Comment · Reviewer_vdWQ · 2024-03-27
> > >
> > > Thank you for addressing my questions and making the source code publicly available! I appreciate your answers.

---

> > > > ### Author Response · Authors · 2024-03-27
> > > > **Thanks to Reviewer vdWQ**
> > > >
> > > > Thank you for your reviews and kind feedback! Your suggestions have helped us improve the quality of our work. We appreciate your time in reviewing our paper.

---

### Meta-Review · Area_Chair_xP5E · 2024-04-04

**Recommendation:** Accept (Poster)
**Confidence:** 5

**Metareview:**

Reviewers' initial critiques were mostly about experimental detail rather than the significance of the proposed method. The authors' feedback addressed those critiques, and all reviewers appreciated the authors' responses and confirmed that they clarified all their questions.

---

### Decision · Program_Chairs · 2024-04-06

Accept (Poster)